# The integrated molecular and histological analysis defines subtypes of esophageal squamous cell carcinoma

Guozhong Jiang[1,15], Zhizhong Wang[2,3,4,5,15], Zhenguo Cheng[3,4,5,15],
Weiwei Wang[1,15], Shuangshuang Lu[3,4,5], Zifang Zhang[3,4,5], Chinedu A. Anene[6,7],
Faraz Khan[6], Yue Chen[6], Emma Bailey[6], Huisha Xu[3,4,5], Yunshu Dong[8],
Peinan Chen[2,3,4,5], Zhongxian Zhang[3,4,5], Dongling Gao[1,3,4,5], Zhimin Wang[3,4,5],
Jinxin Miao[3,4,5], Xia Xue[3,4,5], Pengju Wang[3,4,5], Lirong Zhang[9],
Rathi Gangeswaran[10], Peng Liu[10], Louisa S. Chard Dunmall[10], Junkuo Li[11],
Yongjun Guo[2], Jianzeng Dong[12,13,14], Nicholas R. Lemoine[3,4,5,10],
Wencai Li[1,16] ✉, Jun Wang[6,16] ✉ & Yaohe Wang[3,4,5,10,16] ✉

Esophageal squamous cell carcinoma (ESCC) is highly heterogeneous. Our understanding of full molecular and immune landscape of ESCC remains limited, hindering the development of personalised therapeutic strategies. To address this, we perform genomic-transcriptomic characterizations and AI-aided histopathological image analysis of 120 Chinese ESCC patients. Here we show that ESCC can be categorized into differentiated, metabolic, immunogenic and stemness subtypes based on bulk and single-cell RNA-seq, each exhibiting specific molecular and histopathological features based on an amalgamated deep-learning model. The stemness subgroup with signature genes, such as *WFDC2*, *SFRP1*, *LGR6* and *VWA2*, has the poorest prognosis and is associated with downregulated immune activities, a high frequency of *EP300* mutation/activation, functional mutation enrichment in Wnt signalling and the highest level of intratumoural heterogeneity. The immune profiling by transcriptomics and immunohistochemistry reveals ESCC cells overexpress natural killer cell markers *XCL1* and *CD160* as immune evasion. Strikingly, *XCL1* expression also affects the sensitivity of ESCC cells to common chemotherapy drugs. This study opens avenues for ESCC treatment and provides a valuable public resource to better understand ESCC.

Oesophageal cancer (EC) is an aggressive and invasive disease, associated with one of the highest mortality rates (509,000 per year) and incidence rates (572,000 new cases per year) worldwide in 2018[1]. The global incidence and mortality of EC are predicted to increase in the coming decades[1,2]. The highest prevalence of EC occurs in Asia and Africa, where the most common subtype is oesophageal squamous cell carcinoma (ESCC)[1]. Despite advances in therapeutic options, including

novel targeted therapies and cancer immunotherapies, the prognosis of ESCC remains poor, with a 5-year survival of <15%[3,4]. The major challenges facing ESCC treatment are the aggressive progression and late diagnosis. Therefore, studies on the molecular features of ESCC to identify biomarkers for early diagnosis and key molecular alterations affecting the prognosis of the disease are crucial for early intervention and an improved therapeutic strategy. Several major international

studies have made important advances in identifying the molecular landscapes and understanding the molecular mechanisms of ESCC[5–9]. They highlighted common deregulation of RTK/RAS/PI3K and WNT/Notch pathways, cell cycle regulation, frequently mutated genes such as *TP53*, FAT1, *NOTCH1*, *KMT2D*, *NFE2L2* and *ZNF750*, and epigenetic alterations of ESCC[10]. However, the genetic events associated with the heterogeneous behaviour of ESCC are still poorly understood, resulting in a lack of robust biomarkers for predicting prognosis or designing effective targeted therapeutics. Moreover, the immune landscape and precise immune escape mechanisms of ESCC have not been fully revealed, and there is no effective immunotherapeutic regime available for ESCC even though immunotherapeutic agents will be featured in standard systemic treatment for ESCC in the near future, as it is significantly associated with inflammation and host immunity against dysplastic cells[11]. Therefore, an integrated multi-omics investigation of ESCC to decipher the molecular and immune heterogeneity is required for a thorough understanding of disease pathogenesis, especially for patients from the region with the highest incidence of ESCC.

Here we present a comprehensive genomics and transcriptomics analysis of tumours with matched normal tissue in untreated ESCC patients with more than 4 years of follow-up after surgical resections. We explored the transcriptomic subtypes and diverse immune microenvironments, as well as their prognostic potential, and uncovered tumour-intrinsic immune escape mechanisms. We further identified significant gene and pathway alterations that contributed to the adverse phenotypes. In addition, we developed a deep-learning model to extract and compare subtype-specific histopathological features based on digitised whole-slide histology images (WSI) of the samples. Our study broadens the knowledge of ESCC molecular and histological diversity and provides potential therapeutic targets for the treatment of ESCC.

## Results

### Transcriptomics subtypes of ESCC with distinct prognosis

To fully understand the transcriptomic heterogeneity and molecular signatures of ESCC associated with differences in prognosis, we investigated the transcriptomic landscape of 120 treatment-naive ESCC tumours prospectively collected under strict protocols (Supplementary Data 1). Unsupervised clustering of RNA-seq profiling using non-negative matrix factorisation revealed four stable clusters (Fig. 1a and Supplementary Fig. 1a, Supplementary Data 2). Functional annotation of representative genes in each cluster annotated these subtypes as 'differentiated', 'immunogenic', 'metabolic' and 'stemness' (Fig. 1b, c, Supplementary Data 2 and 3). Gene signatures of these four subtypes were validated in three independent patient cohorts[12–14] (Supplementary Fig. 1b). Keratinocyte differentiation and epidermis development genes such as *LCE3D*, *CDSN* and *KLK5* defined the differentiated subtype. B-cell surface markers *MS4A1*, *CD79A* and *MZB1* and T-cell chemokine ligands *CXCL9*, *CXCL13* and *CXCL10* characterised the immunogenic subtype. The metabolic subtype is associated with the upregulation of genes involved in drug metabolism by cytochrome P450 and retinol metabolism, such as *GSTA1*, *ADH7*, *UGT1A10* and *UGT1A3*. High expression of *WFDC2*, *PEG10*, Wnt signalling modulator *SFRP1* and squamous cell carcinoma (SCC) stem cell marker *LGR6*[15] defined the stemness subtype (Fig. 1a and Supplementary Data 2). All immune-associated pathways, such as the interferon-gamma pathway, TCR pathway and chemokine-signalling pathway, were significantly downregulated in the stemness subtype (Fig. 1b). The transcription factor (TF) profiling further highlighted subtype-specific TFs, including the upregulated activity of *MYB*, *SOX10*, *SP5* and *ARNT2* in the stemness group (Supplementary Fig. 2).

We next utilised a large single-cell RNA-seq data set of samples from 60 individuals with ESCC[16] to validate our subtype specific gene signatures and identify associated cell types in the cancer tissue. Out of

the 208,659 cells, 44,122 were epithelial cells that were dominantly cancer cells (Supplementary Fig. 3), and signature genes from differentiated, metabolic and stemness subtypes were mainly expressed by epithelial cells, while most signature genes of our immunogenic subtype were all expressed by non-tumour cells. For example, *MS4A1*, *CD79A* and *MZB1* were expressed by B cells, *CXCL9* was expressed mainly by myeloid cells, with some in fibroblasts, endothelial and pericytes (Fig. 1a and Supplementary Fig. 3b). To further dissect the heterogeneity of ESCC epithelial cells, the NMF clustering (with $k = 10$ factors) was performed on epithelial single cells to identify diverse transcriptional programmes (Supplementary Fig. 4). Based on shared signature genes and pathway activities, their corresponding programmes of Zhang et al.[16], and transcriptomic subtypes were assigned. Reassuringly, all previous eight expression programmes of epithelial cells[16] were identified in NMF programmes, and these NMF transcriptional programmes seemed to capture all our four transcriptomic subtypes derived from bulk RNA-seq. For example, the NMF cluster 5 and 10 corresponded to our differentiated subtype and the epithelial differentiation (Epi1/2) programme identified by Zhang et al.[16], with the overlap of many signature genes, such as *LGALS7*, *LGALS7B*, *KRT16*, *KRT6B/C*, *FABP5* and *LY6D* of the Epi1 programme, *S100A7/8/9*, *SPRR1A/B* and *SPRR2D* of the Epi2 programme. Our metabolic subtype corresponded to the NMF cluster 4 and the oxidative stress or detoxification (Oxd) programme, with shared genes as *CES1*, *ALDH1A1*, *ALDH3A1*, *AKR1C1/2/3* and *GPX2*. The NMF cluster 6 had the most activated immune and cell adhesion pathways, and shared many mucosal immunity-like (e.g., *S100P*, *CXCL17*, *AGR2* and *MUC20*) and antigen presentation programme genes (e.g., *CD74*, *HLA-DPA1*, *HLA-DRA/B1/B5*, *HLA-A/B/C* and *B2M*). Thus, this cluster mostly likely corresponded to our immunogenic ESCC cells. Interestingly, our stemness subtype was captured by the NMF cluster 1 with many shared genes, such as *SFRP1*, *COL9A3*, *WFDC2* and *LGR6*, although this was not characterised by the eight epithelial programmes of Zhang et al., (Supplementary Fig. 3e and Supplementary Fig. 4). Cluster 1 also had significantly upregulated Wnt signalling and NCAM1 interactions, and the most downregulated keratinisation/cornified envelope and metabolism pathways, which were all signature pathway activities for the stemness subtype. Therefore, the single-cell results further validated our findings derived from bulk tissue RNA-seq and supported our four distinct transcriptomic subtypes.

Importantly, the stemness subtype was associated with the worst overall survival of all subtypes (Fig. 1d, log-rank test $P = 0.028$). The significance as a prognostic biomarker of the top four genes (*WFDC1*, *SFRP1*, *LGR6* and *VWA2)* related to the stemness signature was further proven in an independent cohort of 65 ESCC patients using RT-PCR (Fig. 1e, $P = 0.031$). Subsequently, we investigated the SFRP1 expression in ESCC by Immunohistochemistry (IHC) and Western blot assay. The frequency of SFRP1 protein expression was low in human primary ESCC tumour, showing that SFRP1 protein was positive in 4.3% (3/70) of ESCC tumour tissues and no positive in the matched normal samples, as demonstrated in Supplementary Fig. 5a, b, SFRP1 exhibited positive expression in part clinical specimens and cell lines. Further functional experiments demonstrated that the overexpression of SFPR1 in KYSE-70 and KYSE-140 cell lines could significantly increase cell proliferation in vitro or xenograft tumour growth in vivo, while SFRP1 knockdown in KYSE-450 and KYSE-520 cells exhibited the opposite effects (Fig. 1f and Supplementary Fig. 5c–j).

### Distinct histological features among transcriptomic subtypes

As we also matched Hematoxylin and Eosin (H&E) stained pathology slides for samples that were profiled by sequencing, we next explored if there were any unique histological features specific to each molecular subtype. To quantify and compare the histopathological data/features extracted from the scanned H&E histology images, we developed a deep-learning model using five state-of-the-art

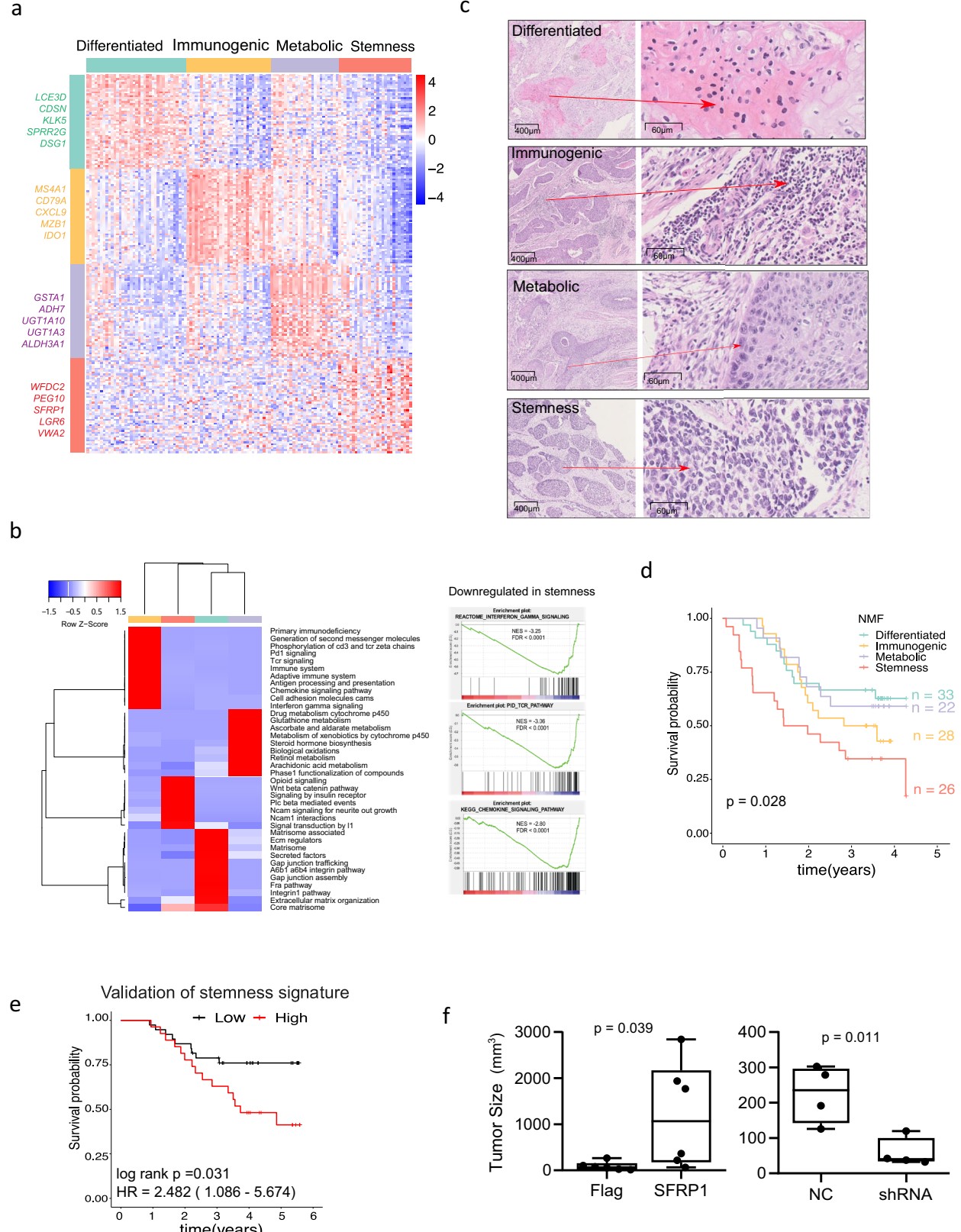

convolutional networks, namely Inception-V3, Inception-ResNet-V2, DenseNet-121, VGG16 and ResNet-50, and performed feature extraction on selected tiles from each whole slide image (WSI) (Supplementary Fig. 6a). This model diversity can enhance performance by capturing different predictive elements and building more enriched representations into the system. This ensemble approach was

reported to retain more informative features for the final retrieval and achieve better accuracy than a separate feature extraction[17–19]. We then compared features among the four transcriptomic groups (Supplementary Data 4), and identified meta features strongly associated with each group (i.e., combined features that were significantly higher in one group compared to the rest) (see Methods, Supplementary Fig. 6,

**Fig. 1 | Transcriptomic subtypes of Chinese ESCC. a** Four distinct transcriptomic subtypes were identified using non-negative matrix factorisation (NMF). The expression heatmap of all representative genes from the four clusters is displayed, and the top five representative genes are shown next to each cluster. Each row represents a representative gene, and each column represents a patient. **b** Heatmap of top enriched pathways for each subtype is shown. Each row represents a significant pathway curated from the mSigDB database (v.6.2). Four subtypes are shown in the column. The '$-\log_{10}$' transformed $p$-values from the hypergeometric test were used to generate this heatmap. Red indicates that the pathway is highly enriched for the gene set. Blue indicates that no enrichment was observed for the gene set. For the stemness subtype, results of three mostly downregulated pathways, 'interferon gamma signalling', 'TCR pathway' and 'chemokine signalling pathway' from GSEA are shown. Normalised enrichment scores (NES) and FDR values are also displayed. **c** Representative histopathology images for the four subtypes are shown. A deep-learning model was developed to extract and compare subtype-specific histological features based on histology slides. The high-magnification pictures were shown with arrows indicating their locations in the

slides in the right panel. These features clearly discriminate the molecular subtypes. **d** A Kaplan–Meier curve is shown comparing patients from the four subtypes with a log-rank $p$-value calculated. **e** A Kaplan–Meier curve is shown for patient samples with high and low stemness signatures in an independent cohort ($n = 63$). The stemness signature was measured as the average expression readout of four genes, *LGR6*, *VWA2*, *WFDC1* and *SFRP1*, by RT-PCR. The patients were split into high and low groups based on an optimal cut-off with R survminer package (see Methods). For all survival curves, significance was determined using a two-sided log-rank test. **f** The effect of SFRP1 overexpression (in KYSE-70, $n = 6$) or knock-down (in KYSE-520, $n = 6$) on the tumour growth of ESCC was evaluated by the tumour growth of SFRP1-modified ESCC cells in immune-deficient mice. All mice in the overexpression group developed tumours, while two mice in the knockdown group had no tumour formation. The tumour size is presented at the end time point of the study (30 days after transplantation of the ESCC cells). The box bounds the interquartile range divided by the median, with the whiskers extending to the min and max values. Significance was determined using a two-sided Wilcoxon test. Source data are provided as a Source Data file.

Supplementary Data 5). Imaging tiles with the highest scores of each feature were selected for review by a pathologist (Fig. 1c). Indeed, tiles with the highest subtype-specific features all contained the distinct histopathological features corresponding to each subgroup (Fig. 1c, Supplementary Fig. 6b). 'DIFF-Feature', characterised by keratin pearls within tumours, is significantly higher in the differentiated samples than in the non-differentiated samples. The 'MET-Feature', marked by eosinophilic cytoplasm with less immune cell infiltration, is higher in metabolic than in the immunogenic, stemness and differentiated groups. By contrast, the 'IMM-Feature' with extensive immune cell infiltration within the stroma of tumours, is higher in the immunogenic than in the non-immunogenic groups. Finally, the STEM-Feature associated with poorly differentiated tumour cells and few immune cells infiltrating within the tumour is the highest in the stemness group compared to other groups (Fig. 1c and Supplementary Fig. 6b, c).

## Tumour immune cell infiltration and tumour cells expressing NK marker genes

We next examined the tumour immune environment of patient tissues based on RNA-seq and determined how this correlated with the four transcriptomic subtypes and prognosis. Several in silico immune deconvolution published methods[20–25] were benchmarked against molecular protein staining of CD4$^+$ and CD8$^+$ cells, tumour cellularity and copy number profiles (Methods, Supplementary Fig. 7), and the best-performing method, the Danaher et al.[22] signature was used to investigate the tumour-infiltrating immune cell populations within the 120 ESCC tumours. Consensus clustering based on the immune cell estimates revealed three distinct clusters: C1 is associated with a generally low level of immune cell infiltration, but with relatively high levels of neutrophils, dendritic and mast cells; C2 is marked by a high level of immune infiltration of B cells, T cells and macrophages; C3 is correlated with a low level of infiltration of all immune cell populations, except for NK cell markers (Fig. 2a). The C3 cluster is significantly associated with high expression of NK cell markers *XCL1*, *XCL2* (Fig. 2b) and *CD160* (Supplementary Fig. 8a).

We further performed IHC of immune cell markers of CD4, CD8 and CD56 (Supplementary Fig. 9), quantified and compared the IHC measurements among the transcriptomic and immune subtypes. The results demonstrated that the immunogenetic subtype and the C2 subtype indeed had the highest level of CD8$^+$ and CD56$^+$ cell infiltration, slightly less so for CD4$^+$ cells (Fig. 2c, Supplementary Fig. 10a, b). It is noteworthy that the C3 cluster had a low level of CD56 by IHC, but high mRNA levels of *XCL1*, *XCL2* and *CD160*. The mRNA expression of other NK markers, such as *NKG7* and *KLRC1*, was the highest in the C2 cluster but low in C3 (Supplementary Fig 8a). This interesting but conflicting observation of NK marker genes warrants further investigation later.

The prognostic values for all profiled immune cells were then assessed in our cohort using a multivariate Cox regression analysis, accounting for patient age, drinking and smoking history, tumour stage and grade, and the higher NK cell abundance was the strongest poor prognostic factor (Fig. 2d, e (China), log-rank test $P = 0.019$). This negative correlation with overall survival for NK cell markers was also seen in the TCGA cohort of 90 ESCC samples (Fig. 2e (TCGA), $P = 0.015$). The comparison between the four transcriptomic subtypes and three immune profile clusters demonstrated a non-random correlation (Fisher's exact test, $p = 9.37e{-}11$). Fifteen of 31 cases (48.4%) from the C3 cluster were stemness tumours, while the differentiated and immunogenic subtypes were overrepresented in the C1 and C2 clusters, respectively (Fig. 2f). Of note, we also observed positive correlations of mRNA expression between *LGR6* and *XCL1* ($r = 0.40$, $p < 0.0001$), *XCL2* ($r = 0.39$, $p < 0.0001$) and *CD160* ($r = 0.32$, $p = 0.0003$) (Fig. 2g), suggesting a degree of certain association between stemness and NK cell estimates.

To further investigate this correlation, IHC staining for XCL1, CD160 and LGR6 was performed in the matched serial sections of tumour tissues in order to determine the spatial composition and cell types that expressed these markers (Supplementary Fig. 11). Surprisingly, XCL1 and CD160 were predominantly expressed in tumour cells (Fig. 2h), with a few positive immune cells infiltrated in the stroma. We also observed a positive correlation between *XCL1/2* gene expression and tumour cellularity based on sequencing data (Supplementary Fig. 8b). XCL1 or CD160 expression co-localised with LGR6 expression (Fig. 2h, Supplementary Fig. 8c), and co-expression within tumours was seen in 16.3% (16/98) of our cohort for XCL1 and LGR6, and 27.8% (25/90) for CD160 and LGR6 (Supplementary Data 6 and 7). When assessing all 98 samples with available IHC, the LGR6 and XCL1 IHC staining appeared to be significantly associated (co-expressed, two-tailed Fisher's exact test, $P < 0.0001$), while assessing all 90 samples with IHC, LGR6 and CD160 staining was not significantly associated ($P = 0.93$, Supplementary Data 7).

Of note, XCL1 was predominantly expressed in cancer cells showing adenocarcinoma morphology and dysplastic cells in the submucosa glands, while CD160 could be expressed in both squamous carcinoma and adenocarcinoma cells (Fig. 2h, Supplementary Fig. 8c) as well as in the proliferative and dysplastic cells of the submucosa glands (Supplementary Figs. 8d and 11). Interestingly, we observed that XCL1-expressing dysplastic cells in the submucosal gland in most of the cases are completely separated from the squamous cell carcinoma, suggesting that this subgroup of patients might concurrently have both squamous cell carcinoma and adenocarcinoma, or this adenosquamous carcinoma might be derived from submucosa gland epithelial cells. Our finding of tumour cells overexpressing NK cell markers *XCL1/2* and *CD160* correlating with the lowest level of immune

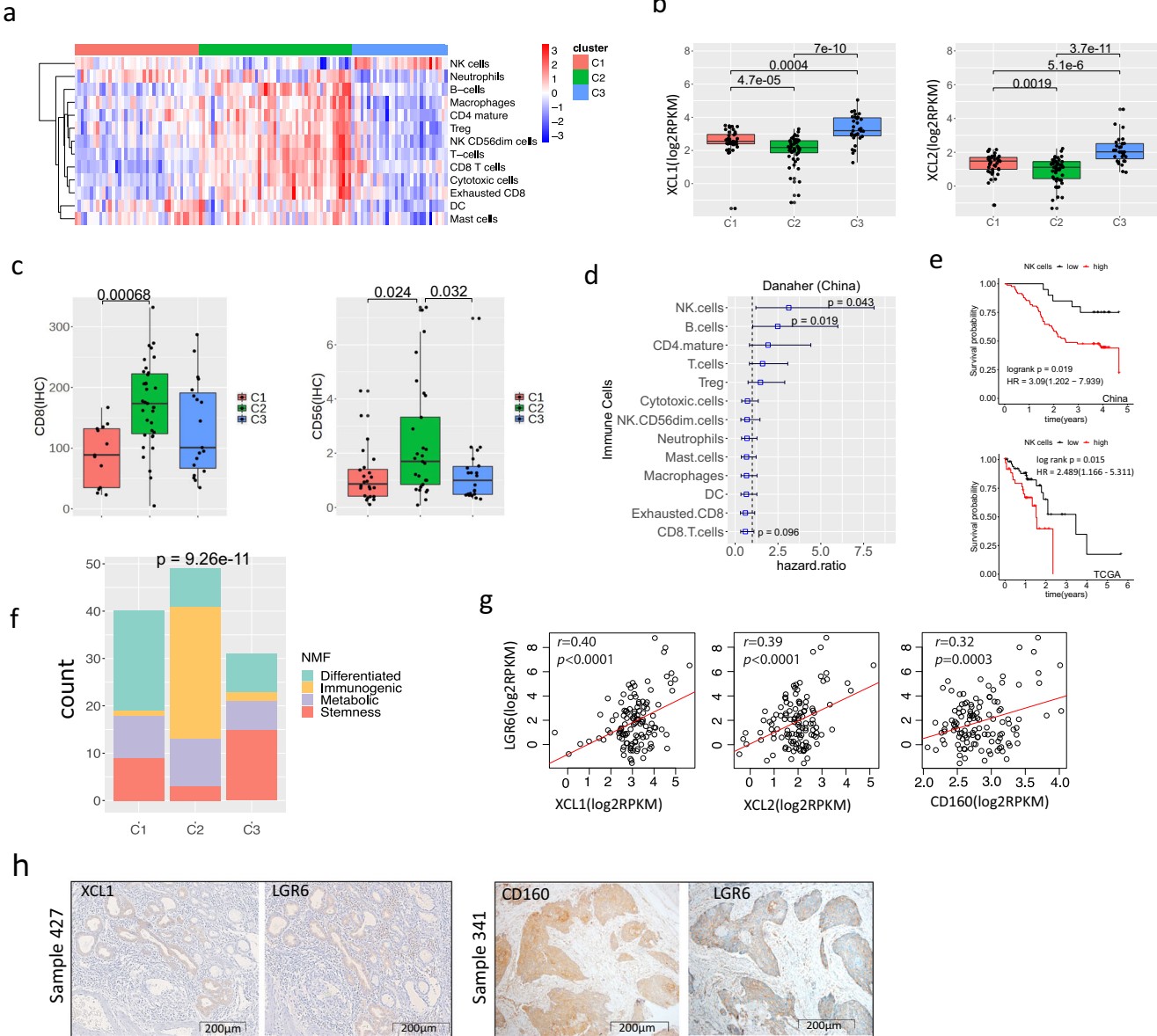

**Fig. 2 | Heterogeneity of immune cell infiltration in ESCC. a** The immune cell infiltration profiling across our cohort is shown, clustered by the level of estimated immune cell infiltration. Each row represents an immune cell type as estimated by the method used by Danaher et al. Immune cells are natural killer (NK) cells, neutrophils, B cells, macrophages, CD4+ mature cells, regulatory T cells (Treg), CD56dim NK cells, total T cells (T cells), CD8+ T cells, cytotoxic cells, exhausted CD8+ T cells (exhausted CD8), dendritic cells (DC) and mast cells. Consensus clustering was performed. Each column represents a patient sample. Three immune infiltration clusters were identified: C1, C2 and C3. **b** Levels of gene expression of *XCL1* and *XCL2* for n = 120 samples are shown among the three immune subtypes as a box and whisker plot. Significance in each pairwise comparison is shown using the two-sided Wilcoxon rank-sum test. **c** IHC analysis revealed that the C2 immune subtype had significantly increased levels of CD8 (67 samples) and CD56 (75 samples) expression. Significance was determined using a two-sided Wilcoxon test; *p < 0.05, ***p < 0.001. **d** The survival analysis of all profiled immune cell types against overall survival for 102 samples is shown. The hazard ratio (HR) derived from the multivariate Cox regression model is shown as a whisker plot. The blue square indicates the HR value, and the error bars represent 95% confidence intervals. Significance is determined using a two-sided log-rank test (■ p < 0.1;

* p < 0.05). **e** A Kaplan–Meier curve is shown for NK cell estimates against overall survival for our cohort (China, 102 samples) and TCGA (90 samples). Multivariant survival analysis was performed for the China cohort. HR and *p*-value derived from the log-rank test are shown. **f** The number of cases of the four transcriptomic subtypes is shown among the three immune subtypes C1, C2 and C3. Fisher´s exact test was used to test if there is any difference in the proportion of transcriptomic subtypes between different immune subtypes (**** p < 0.0001). **g** The scatter plot of expression levels between *LGR6* and three NK cell markers, *XCL1*, *XCL2* and *CD160*, is shown. Two-sided Pearson's correlation coefficient and associated *p*-value are displayed. **h** IHC (Immunohistochemistry) staining of *XCL1* and *LGR6* from one patient, Sample 427, and IHC of *CD160* and *LGR6* from a different patient, Sample 341, are shown. The IHC results show that *XCL1* and *LGR6*, *CD160* and *LGR6* are co-expressed in tumour cells. Furthermore, to provide a more comprehensive understanding of our findings, we included a larger visualisation of IHC results depicting CD160, LGR6, XCL1, and CD56 in both normal control and tumour samples for Sample 333 in Supplementary Fig. 11a. In **b, c**, the box bounds the interquartile range divided by the median, with the whiskers extending to a maximum of 1.5 times the interquartile range beyond the box. Source data are provided as a Source Data file.

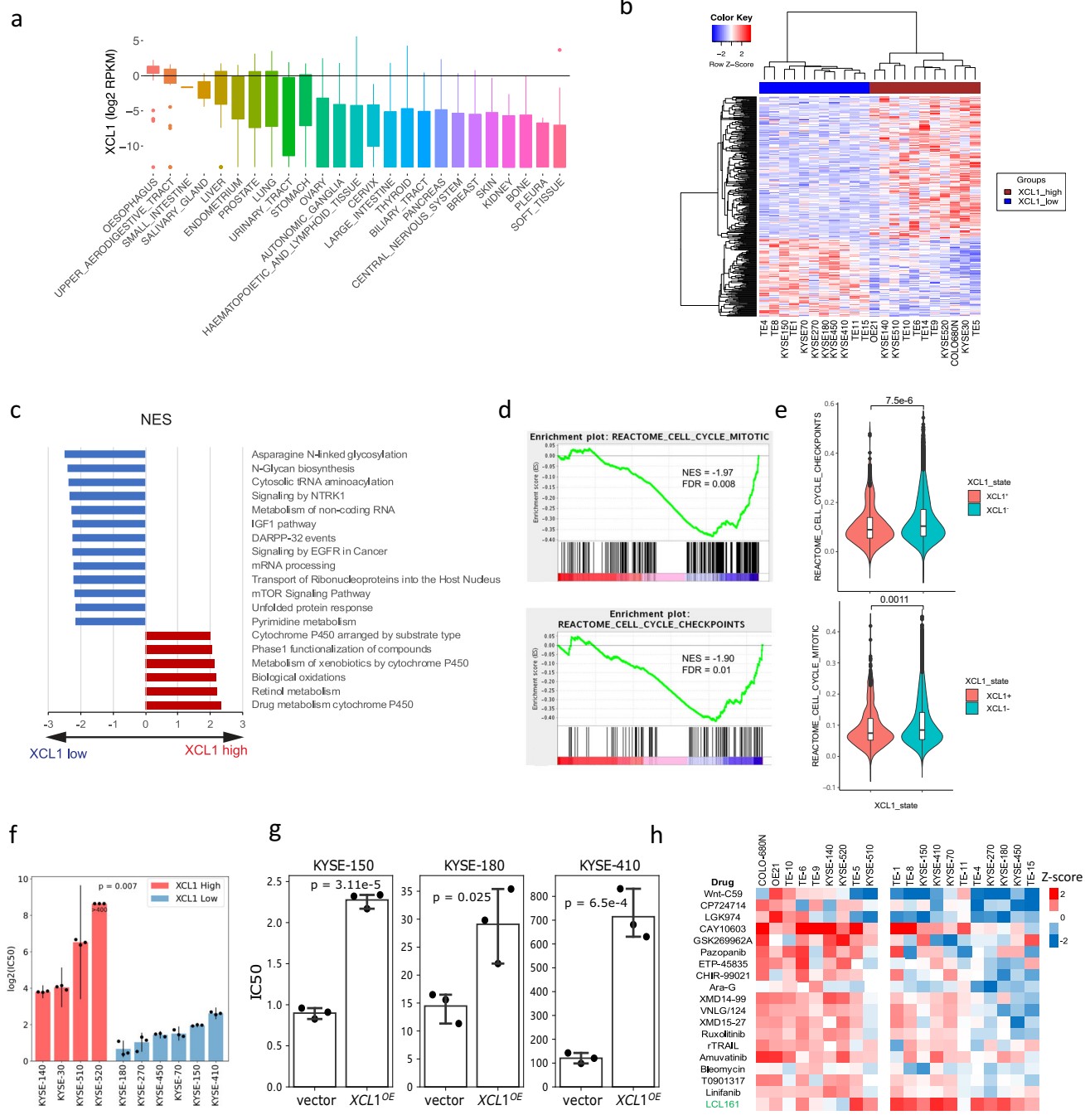

**Fig. 3 | Characteristics of XCL1-high ESCC cells. a** The expression of *XCL1* across all 1,019 profiled cell lines in the Cancer Cell Line Encyclopaedia (CCLE) is shown. The expression was measured in log₂ transformed RPKM. The whiskers extend to a maximum of 1.5 times the interquartile range beyond the box. **b** The heatmap of significantly differentially expressed genes between ESCC cells of *XCL1* high and low groups is shown. **c** The top upregulated and downregulated pathways (GSEA) derived from the *XCL1* high and low differential expression analysis is shown. The pathways were sorted based on the normalised enrichment score (NES). **d** The cell cycle gene sets were downregulated in *XCL1*-high cells compared to *XCL1*-low cells based on GSEA of RNA-seq data. **e** The violin plots depict the distribution of cell cycle gene set enrichment levels between 515 *XCL1*-positive cells and 32,944 *XCL1*-negative cells obtained from single-cell data collected by Zhang et al.[16]. The levels of cell cycle activity were compared between these two groups using Wilcoxon rank-sum test (**P < 0.01, ****P < 0.0001). The inset box bounds the interquartile range divided by the median, with the whiskers extending to a maximum of 1.5 times the

interquartile range beyond the box. **f** The cytotoxicity of 5-FU in a panel of human ESCC cell lines is shown. The cells are divided into *XCL1* high (red) and low (blue) groups based on the CCLE separation. For each cell line we conducted three repeated experiments to determine the IC50, here log₂ transformed mean IC50 scores and standard deviation are shown, the *P* value was calculated using two-sided Mann Whitney test. **g** The mean IC50 value of 5-FU and their standard deviation derived from three repeat experiments between control and *XCL1* over-expressing cells of KYSE-150 (*P* = 3.11e−5), KYSE-180 (*P* = 6.5e−4) and KYSE-410 (*P* = 0.025) is shown. The IC50 difference was compared using a two-sided *t*-test. **h** The drug screening profiling between ESCC *XCL1*-high and low cells is shown, based on data generated by the Genomics of Drug Sensitivity in Cancer (GDSC) resource. The drugs that show significant differences in IC50 (Student's *t*-test, *P* < 0.05) between the two groups are selected. High and low levels of resistance are indicated in red and blue, respectively. Source data are provided as a Source Data file.

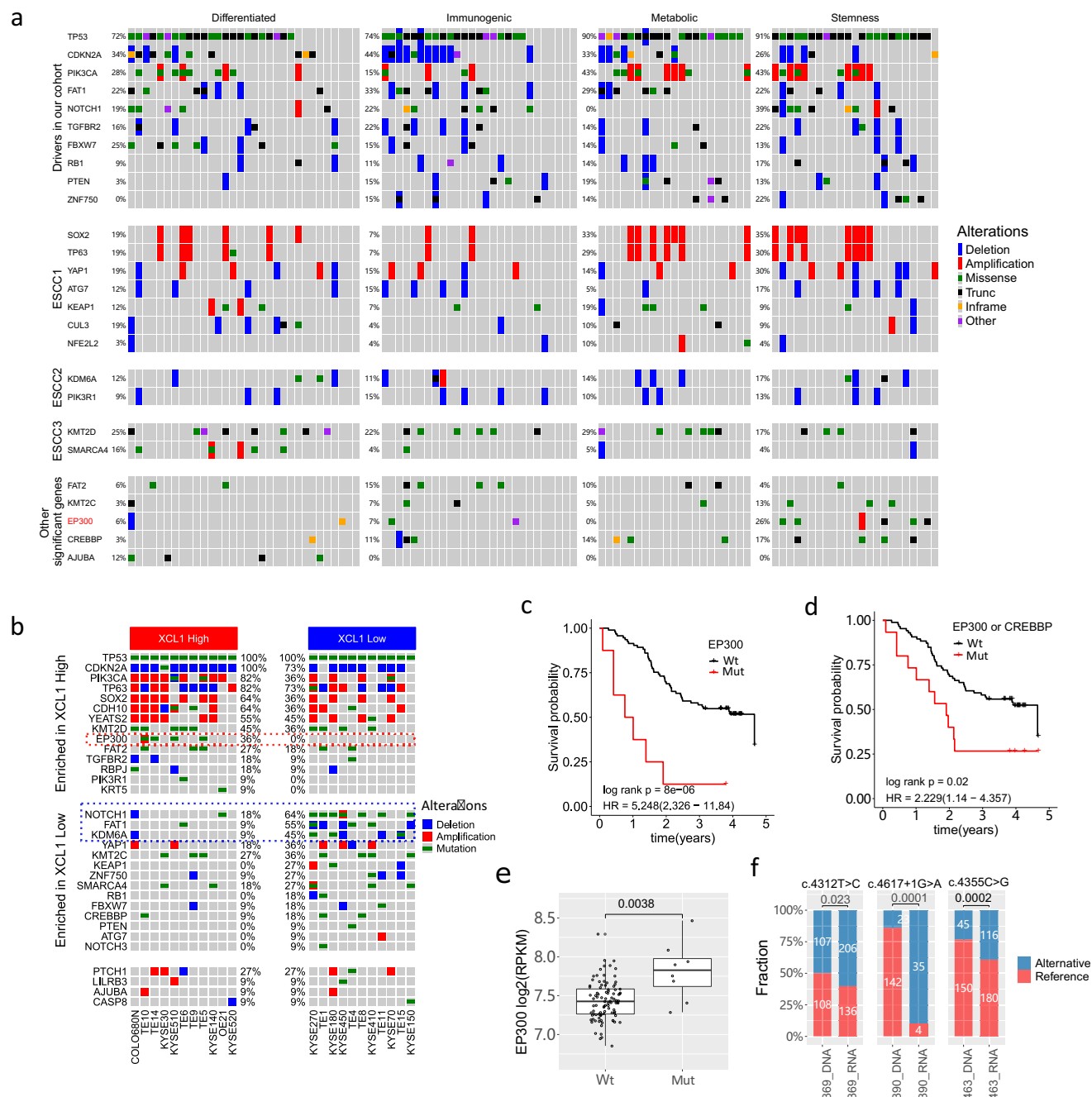

**Fig. 4 | The landscape of genomic alterations in ESCC. a** The genomic landscape of ESCC driver genes and previously reported ESCC subtype-specific genes is shown among four transcriptomic subtypes in our cohort. ESCC1/2/3 denotes the ESCC subtypes identified by the TCGA 2017 EC study. Additional significant genes reported from previous large Chinese ESCC cohort studies were also included. For copy number aberrations, only amplifications and deletions were included. Cases of copy gain or loss were not counted. **b** The genomic landscape of significant ESCC genes across *XCL1*-high and low ESCC cell lines. **c** The overall survival of *EP300*-mutated (Mut) versus wildtype (Wt) cases. **d** The overall survival of *EP300* and/or *CREBBP* mutated (Mut) versus wildtype (Wt) samples. **e** The EP300 gene expression levels for 102 samples were compared between EP300-mutated and wild-type samples using the Wilcoxon rank sum test to assess the significance of expression differences between groups. (**$P < 0.01$). Two cases with amplification or deletion only were excluded. The box bounds the interquartile range divided by the median, with the whiskers extending to a maximum of 1.5 times the interquartile range beyond the box. **f** Reference and alternative allele counts and percentages between DNA WES and RNA-seq for three *EP300*-mutated cases, 369, 390 and 463, with missense and splice site mutations. Fisher's exact test was performed to determine the allelic imbalance between DNA and RNA, *$P < 0.05$, ***$P < 0.001$. Source data are provided as a Source Data file.

cell infiltration suggests a tumour-intrinsic immune evasion mechanism of ESCC. Furthermore, we also found that both *XCL1* and *XCL2* had much more elevated expression in patient tumour samples compared to their matched normal ($P < 0.0001$), and within tumour samples, high *XCL1* expression was significantly associated with worse overall survival (Supplementary Fig. 12). All these observations suggest that

XCL1 expressed by tumour or dysplastic cells may have a tumour-promoting role in ESCC.

## Molecular characteristics of *XCL1*-high ESCC tumour cells

Characterisation of tumour cells expressing NK marker genes, especially *XCL1*, is essential to uncover molecular mechanisms and

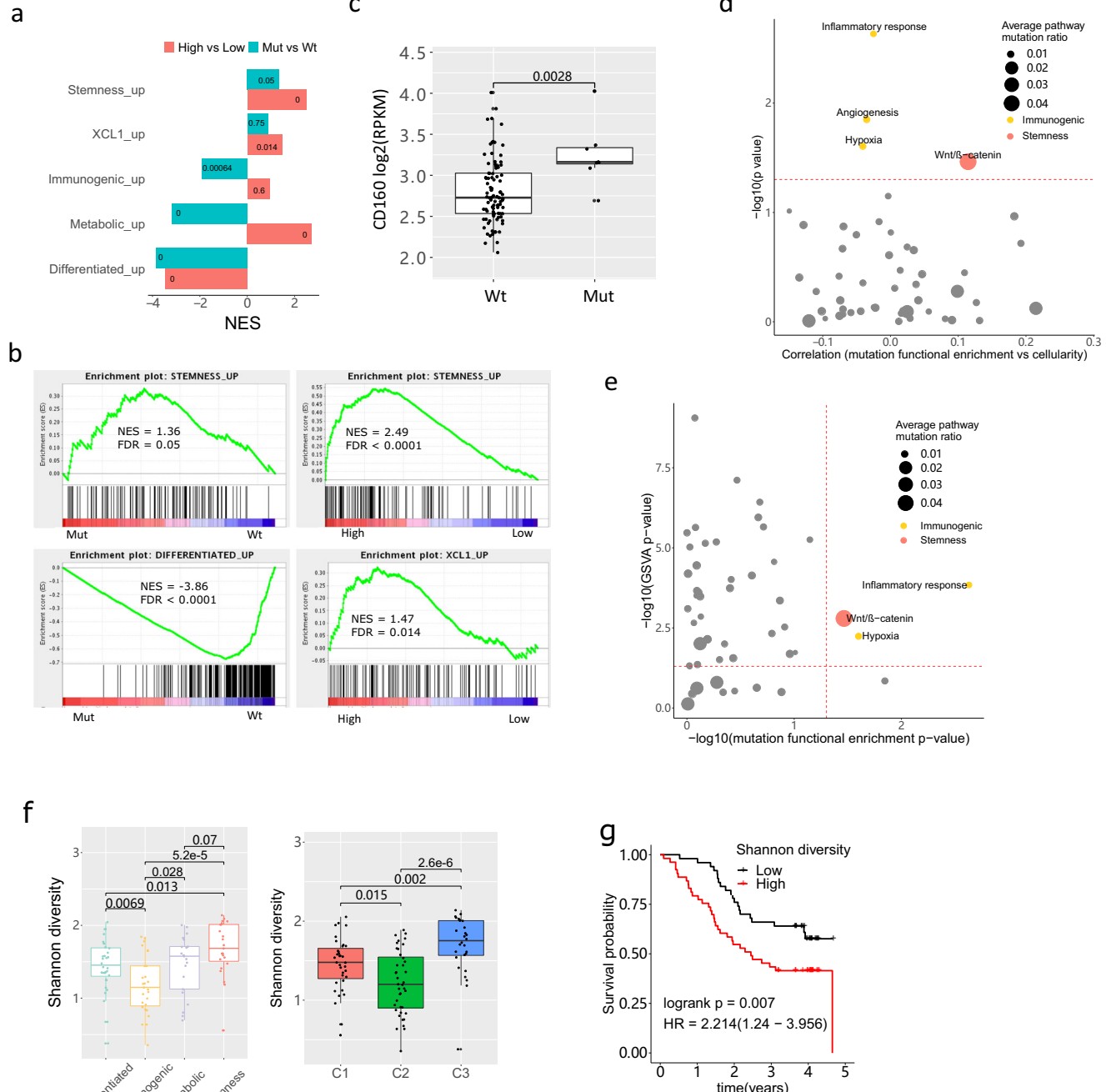

**Fig. 5 | EP300 mutations, overexpression in ESCC and pathway functional mutation enrichment. a** GSEA normalised enrichment score (NES) of *EP300 Mut* (mutated) versus *Wt* (wildtype) samples (green bars) and *EP300* high versus low expression samples (red bars), against representative (upregulated) gene sets of four transcriptomic subtypes and upregulated genes in *XCL*-high ESCC cell lines (XCL_up). The values of FDR were shown within the bars. **b** GSEA plots of 'stemness_up' and 'differentiated_up' gene sets for the *EP300 Mut* (mutated) versus *Wt* (wildtype) comparison, and the 'stemness_up' and 'XCL1_up' gene sets for the *EP300* high versus low-expression comparison. FDR *q*-values were shown. **c** Gene expression of *CD160* between *EP300* mutated and wildtype samples. Wilcoxon rank-sum test was used to compare the level between groups. **d** Pathway functional mutation enrichment adjusted for tumour mutation burden and the correlation between functional mutation enrichment ratio and tumour cellularity are plotted together for each hallmark gene set. *P*-values derived from the Kruskal–Wallis test comparing the enrichment ratio among the four subtypes were used for *y*-axis values. For significant hallmark gene sets, they were coloured to represent the

subtype which had the highest enrichment for this gene set. **e** Significance *P*-values (−log10 transformed) comparing the enrichment scores among the four subtypes against *P*-values comparing GSVA values among the four subgroups are plotted for each hallmark gene set. For significant hallmark gene sets that passed both significance thresholds (*P* < 0.05), they were coloured to represent the subtype which had the highest enrichment for that gene set in both functional mutation and pathway expression levels. **f** The intra-tumour heterogeneity for 102 samples, measured as the Shannon density, is shown across the four transcriptomic and immune subtypes, using the Wilcoxon rank-sum test. **g** The survival analysis of the Shannon density index against overall survival is shown using a multivariate analysis. The *P*-value derived from the log-rank test was shown, along with the hazard ratio and 95% confidence interval. All significance is shown in the figure, \**P* < 0.05, \*\**P* < 0.01, and \*\*\*\**P* < 0.0001. In **c**, **f**, the box bounds the interquartile range divided by the median, with the whiskers extending to a maximum of 1.5 times the interquartile range beyond the box. Source data are provided as a Source Data file.

pathways which may offer alternative therapeutic targets and candidate drugs for this subgroup of ESCC. We first utilised the Cancer Cell Line Encyclopaedia (CCLE) RNA-seq data to analyse the association of XCL1/2 with ESCC. Interestingly, EC cell lines had the highest level of XCL1 mRNA expression out of all cancer types profiled (Fig. 3a). Hierarchical clustering based on differentially expressed genes ($n = 413$, $P < 0.01$) between XCL1 high ($n = 11$) and low ($n = 11$) ESCC cell lines clearly separated the two groups (Fig. 3b, Supplementary Data 8). The GSEA results further showed that XCL1-high cells exhibited upregulation of drug metabolism cytochrome P450, retinol metabolism and biological oxidation pathways (false discovery rate, FDR < 0.0001), while asparagine N-linked glycosylation, N-Glycan biosynthesis, cytosolic tRNA aminoacylation and signalling by NTRK1 were highly overrepresented in XCL1-low cells (Fig. 3c). It is worth noting that different subsets of genes contributed to the upregulation of drug metabolism by cytochrome P450 and retinol metabolism were seen in XCL-high cells and the metabolic subtype (Supplementary Fig. 13). We also observed a significantly positive correlation in mRNA expression between XCL1 and LGR6 for ESCC cell lines (Supplementary Fig. 12d, Pearson's correlation $r = 0.59$, $p = 0.004$), further highlighting the association between NK marker XCL1 and the stemness signature in ESCC cell lines. Interestingly, XCL1-high cells also showed significantly downregulated cell cycle gene set enrichment scores compared to XCL1-low cells (Fig. 3d). This was further seen in the single cell data[16], which showed that XCL1 positive epithelial cells ($n = 515$) had significantly lower cell cycle gene set enrichment scores than XCL1 negative epithelial cells ($n = 32,944$) (Fig. 3e, Wilcoxon rank test, $P = 7.48e-06$ for cell cycle checkpoints; $P = 0.001$ for cell cycle mitotic). Although mRNA expression levels of cell cycle-related genes were reduced in XCL1-high cells compared to XCL1-low cells, XCL1 overexpression in ESCC cells did not functionally affect the cell cycle (Supplementary Fig. 14). More work is needed to further elucidate its role in the cell cycle and other pathways associated with ESCC.

Given the upregulation of drug metabolism cytochrome P450 in XCL1-high ESCC cells, we next explored the drug sensitivity to fluorouracil (5-FU), the first-line chemotherapy drug for ESCC, between XCL1-high and -low cell lines. All XCL1-high cell lines tested exhibited a much higher IC50 than XCL1-low cells (Fig. 3f, Wilcoxon rank test, $P = 0.007$). Impressively, over-expression of XCL1 in XCL1-low cell lines, KYSE-150, KYSE-180 and KYSE-410, resulted in a significant increase in IC50 of 5-FU compared to their control cells (Fig. 3g, Supplementary Fig. 15a, b) while overexpression of XCL1 did not affect the cell proliferation (Supplementary Fig. 15d). We then explored the cancer drug sensitivity data of 367 compounds from the Genomics of Drug Sensitivity in Cancer (GDSC) resource[26]. Compared to XCL1-low cells, XCL1-high cell lines displayed a higher level of resistance to almost all drugs screened (Fig. 3h and Supplementary Data 9, IC50 z-score t-test $p < 0.05$). XCL1-low cell lines exhibited a significantly higher sensitivity to Wnt-C59, LGK974 (both targeting PORCN and WNT signalling) and CP724714 (targeting ERBB2 and RTK signalling). Deregulated pathways associated with XCL1-high cells may represent therapeutic targets for this subtype of ESCC with poor prognosis.

## The genomic landscape among molecule subtypes of ESCC

Given the heterogeneous transcriptomic and immune signatures observed in our cohort, we next examined whether the genomic landscapes differ among ESCC subtypes. Whole-exome sequencing of 103 samples with matched RNA-seq was performed and analysed (Supplementary Data 10). We identified, on average, 358 somatic mutations (range 21–3583) and 152 non-silent mutations (range 9–1242) per sample, respectively. Overall and subclonal mutation burden and tumour purity were not correlated (Supplementary Fig. 16a). The overall mutational load, the number of non-silent mutations, and somatic copy number profiling were similar among the four ESCC subtypes (Supplementary Fig. 16b). Three driver gene

detection methods, MutsigCV[27,28], dNdScv[29], OncodriveFM[30], were applied, and 10 ESCC driver genes were identified by at least two methods (Supplementary Fig. 16c), which had all been detected in ESCC previously[5,7–9,31,32]. Incorporating representative genomic alterations of the three molecular subtypes ESCC1/2/3 previously identified from 90 TCGA ESCCs[13] and driver genes that have been implicated in a large series of Chinese ESCCs[5,32], we found that although the alteration frequencies (i.e., mutations, amplifications and deletions) of most significant genes in ESCC were similar among our subtypes, the stemness type had a higher frequency of NOTCH1 and EP300 alterations, affecting 39% and 26% of stemness tumours, respectively (Fig. 4a, Fisher's exact test, $P = 0.018$ and 0.008 for NOTCH1 and EP300, respectively). Interestingly, no alterations in NOTCH1 and EP300 were detected in the metabolic subtype, and no ZNF750 alterations were detected in the differentiated subtype. Stemness tumours seemed to have the highest overall frequencies of ESCC1 and ESCC2 alterations (74% and 65% of cases, respectively). Across our cohort, although the highest overall frequency was observed for ESCC1 alterations (61%), other subtype alterations also occurred substantially, 50% for ESCC2 and 52% for ESCC3 (Supplementary Fig. 16d).

Next, we investigated the genomic alterations of ESCC functional genes identified above across ESCC cell lines. Interestingly, in XCL1-high cells, alterations in EP300 occurred in 4/11 (36%) cases, compared to zero cases in XCL1-low cells (Fig. 4b), corresponding to our result of the highest EP300 alternation frequency in the stemness group. Whereas alterations in NOTCH1, FAT1 and KDM6A seemed to be more prevalent, 64%, 55% and 45%, in XCL1-low cells, respectively, than in XCL-high cells, 18%, 9% and 9%, respectively (Fig. 4b). EP300 is a histone acetyltransferase, and its mutation has been shown to be associated with poor outcome in ESCC and HNSCC[33,34]. We also observed significantly poorer overall survival for patients with EP300 mutations in our cohort (Fig. 4c, HR = 5.248, $P = 8e-06$). When combined with CREBBP mutations, the EP300/CREBBP mutation status was still predictive of worse survival (Fig. 4d, HR = 2.229, $P = 0.02$). Although CREBBP mutations were not enriched in the stemness subtype, alterations in EP300 and CREBBP tended to co-occur ($n = 4$) only in stemness samples (Fig. 4a). Interestingly, EP300-mutated patient samples ($n = 8$, 7.8%) had significantly higher EP300 expression levels than EP300-wildtype samples (Fig. 4e, Wilcoxon rank test, $P < 0.01$). Out of eight EP300 mutations, three single nucleotide mutations (2 missense mutations, c.4312 T > C, c.4355 C > G and one splice site mutation, c.4617+1 G > A) showed significantly elevated levels of the alternative (mutated) allele in RNA- compared to the DNA-level (Fig. 4f). While for two codon-affecting mutations, one nonsense (c.3244 C > T) and one frame-shift deletion (c.1914_1915del), the opposite pattern was observed, with decreased levels of the alternative allele in RNA than in DNA (Supplementary Data 11).

## EP300 mutations and overexpression and their relationship with stemness and NK marker genes in ESCC

We next examined whether EP300 mutations and expression had any associations with stemness and NK marker genes. We first performed differential expression analysis between EP300-mutated and wildtype samples, followed by GSEA against representative (i.e., upregulated) gene sets of four transcriptomic subtypes and XCL1-high genes derived from cell lines. We found that the gene set representative of stemness subtype was upregulated in EP300-mutated compared to wildtype samples (NES = 1.36, FDR = 0.05), while all gene sets of other three subtypes were massively downregulated in EP300-mutated samples (Fig. 5a, b, FDR < 0.001, Supplementary Fig. 17a). Although there was no upregulation of XCL1-high genes, we observed significantly increased gene expression of CD160 in EP300-mutated samples (Fig. 5c, $P = 0.003$). Interestingly, the EP300 gene expression was also the highest in the stemness subtype, and there was a positive

correlation in expression between *EP300*, *XCL1/2* and *CD160* (Supplementary Fig. 17b, 17c).

Next, focusing on *EP300*-wildtype samples, we selected the top 20 *EP300*-high and 20 *EP300*-low expression samples, and found that the stemness gene set and *XCL1*-high genes were significantly upregulated in the *EP300*-high group (Fig. 5a, b, 'stemness up' gene set, NES = 2.49, FDR < 0.0001; *XCL1*-high gene set, NES = 1.47, FDR = 0.014), whereas the differentiated gene set was greatly downregulated (NES = −3.47, FDR < 0.0001). Interestingly, we found that the metabolic gene set was significantly upregulated in *EP300*-high samples (NES = 2.71, FDR < 0.0001), while the immunogenic gene set was not notably affected by *EP300* expression levels (Fig. 5a). Taking all these together, our results demonstrate a strong positive association between *EP300* mutations, overexpression and stemness/NK marker *XCL1* related signatures, suggesting that *EP300* may have a role promoting tumours to become a more aggressive subtype in ESCC.

### Functional mutation enrichment in pathways among ESCC subtypes

To further investigate whether specific pathway-related mutations were associated with transcriptomic subtypes, we performed functional mutation enrichment analysis adjusted for sample mutation load using the 50 cancer hallmark gene sets (see Methods). We identified functionally relevant mutations in Wnt/β-catenin signalling which were significantly more enriched in the stemness samples, while functional mutations in inflammatory response, angiogenesis and hypoxia were highly enriched in the immunogenic tumours (Fig. 5d, Kruskal–Wallis test, $P < 0.05$, Supplementary Fig. 18a). Of note, the hallmark gene set expression profile using GSVA also confirmed the functional differences identified among the four subtypes (Supplementary Fig. 18b). We then evaluated whether the pathway functional mutation enrichment affected the expression of pathway genes, and found that mutation enrichment in Wnt/β-catenin signalling, inflammatory response and hypoxia were positively correlated with pathway gene expression activities for the corresponding ESCC subtypes (Fig. 5e), suggesting that high level of Wnt signalling expression in the stemness group could be a consequence of functional mutations in regulators of the Wnt pathway, and these functional mutations were the most enriched in stemness samples. This is also the case for the high pathway activity of inflammatory response and hypoxia for the immunogenic subtype. Interestingly, the stemness subtype also seemed to have the highest proportion of nonsense mutations in Wnt pathway genes. The missense mutations in the Wnt signalling pathway were highly deleterious, as predicted by SIFT[35] and PolyPhen-2[36], and the stemness group seemed to have the highest proportion of deleterious Wnt mutations (93%), significantly higher than that of the differentiated subtype (60%, Fisher's exact test, $P = 0.035$, Supplementary Fig. 18c).

### Clonality analysis among molecule subtypes of ESCC

Given the well-documented interplay between immune infiltration and tumour clonal evolution[37,38], we finally investigated the level of intra-tumoural heterogeneity (ITH, measured by the Shannon diversity index) based on variant allele frequencies of all mutations in each tumour among the immune and transcriptomic subtypes. Our results showed the greatest ITH in the C3 cluster, whereas in C2 where the immune infiltration was the highest, the diversity was greatly reduced (Fig. 5f). In addition, the highest ITH presented in the stemness subtype while the immunogenic subtype had the lowest ITH (Fig. 5f). The negative correlation between ITH and immune infiltration was further confirmed by IHC of CD8 and CD56 (Supplementary Fig. 10c). Our results suggest that the immune microenvironment strongly influences the tumour evolution and (sub)clonal architecture. The stemness/C3 group with the highest level of immune exclusion also had the highest tumour (sub)clonal diversity. Importantly, high ITH was also a

significant marker of poor prognosis in ESCC (Fig. 5g, multivariate analysis, hazard ratio HR = 2.214, log-rank $P = 0.007$).

## Discussion

This study carried out the comprehensive genomic-transcriptomic characterisation of a large cohort of treatment-naïve ESCC patients with long-term follow-up data from high-incidence areas. Here we performed a thorough integrated analysis of genetic alterations, gene expression and immune cell infiltration, and examined their correlations with clinical and pathological data. We validated our results using independent datasets, and in vitro and in vivo biological experiments. Importantly, we developed a deep-learning model to extract and measure the level of subtype-specific histopathological features based on digitised WSIs among patient samples. Our study offers important insights into genetic events that drive ESCC diversity and progression and reveals a mechanism of ESCC immune evasion.

Firstly, four distinct subtypes of ESCC (differentiated, immunogenic, metabolic and stemness) have been identified and validated in our study, and each subtype shows a unique molecular signature and histopathological changes. Previously, the integrated genomic characterisation of EC had divided ESCC into three molecular subtypes[13,39]. These ESCC subtypes showed trends for geographic associations, while their correlations with clinical outcomes were not demonstrated. The limitation also lies in the fact that the cases originated from the regions with moderate or lower ESCC incidence. Upon investigating their subtype-specific alterations in our cohort and subtypes, we found that the overall frequencies were high and comparable for all three previous subtypes, although stemness tumours seemed to have the highest frequencies of ESCC1 and ESCC2 alterations. The ESCC samples of our study were from the high-incidence populations of China, where ESCC accounts for the vast majority of EC (>90%). Identification of robust biomarkers in these ESCCs has tremendous clinical implications, given that 70% of global EC cases occur in China. Indeed, we found that the stemness subgroup has the poorest prognosis compared to other subgroups. We functionally validated the role of one of the top stemness genes, *SFRP1* (a WNT signalling modulator), in the progression of ESCC. We proved that *SFRP1* significantly enhances the malignant phenotypes of ESCC cells in vitro and progression in vivo, suggesting that targeting the WNT pathway via *SFRP1* may be a promising strategy for the treatment of ESCC as ESCC has a high frequency (up to 86%) of alterations in the WNT pathway[7].

Immune evasion is a hallmark of cancer[40]. Cancer immunotherapy has revolutionised the profile of cancer therapies over the past decade. While long-term survival is observed for a fraction of cancer patients, the majority of patients, including EC patients, currently do not benefit from immunotherapy treatments such as immune checkpoint blockade therapy[11], emphasising the need to identify the genomic and molecular determinants underpinning immune evasion of ESCC. Here we identified ESCC cells overexpressing natural killer (NK) cell markers such as XCL1/2 and CD160. The overexpression of these markers is significantly associated with a shorter overall survival of ESCC patients. XCL1 is a C-class chemokine and is produced predominantly by NK and activated CD8+ T cells[41,42], and the XCL1-XCR1 axis normally plays a crucial role in the induction of effective cytotoxic immunity[43]. A recent study demonstrated that XCL1 expression correlates with CD8-positive T cell infiltration and PD-L1 expression in mature ovarian cystic teratomas, but there is no correlation between XCL1 expression and prognosis or clinical stage[44]. However, in our samples, the *XCL1*-overexpressing tumours showed the lowest immune cell infiltration. Interestingly, the Human Protein Atlas data also provided evidence of *XCL1*-overexpressing tumour cells in a number of cancer types, and this overexpression is significantly correlated with a shorter overall survival rate in patients with colorectal and renal cancer (Supplementary Fig. 15c). The immunogenic role of XCL1 and its underlying functional mechanisms in the development of

ESCC and other tumours need further investigation. CD160, a glycosylphosphatidylinositol-anchored Ig domain protein that is expressed on NK cells, γδ T cells and a minor subset of CD4[+] and CD8[+] T cells, is also overexpressed in some ESCC tumours in our study. Previous studies demonstrated that CD160 is dramatically increased in B cell malignancies[45,46]. CD160 can function as both a co-activating and co-inhibitory receptor, depending on which receptor/ligand is operating in the context of neighbouring interactions[47]. Given the negative correlation of CD160 expression with clinical prognosis in ESCC and other tumours such as renal cancer (Supplementary Fig. 8e), we postulate that CD160-expressing cancer cells could use the HVEM and BTLA inhibitory pathways to inhibit T and NK cells activities[47,48]. These interesting preliminary data warrant further investigation into the detailed pathways of how CD160 expression in cancer cells modulates host immunity. The interactions of CD160 with its ligands may be important in the pathophysiology of ESCC, offering targets for therapeutic manipulation.

Recent studies have shown that cancer stem cells (CSCs) have the ability to hide from the immune system ab initio, evading the immunosurveillance phase[49,50]. Interestingly, we found that in the C3 immune cluster of ESCC characterised by relatively higher levels of *XCL1* and *CD160*, ~50% of tumours are from the stemness subtype, which presents a significant WNT alteration. A recent study demonstrated that latency-competent cancer (LCC) cells from early-stage human lung and breast carcinoma can enter a quiescence state through the expression of the WNT inhibitor DKK1 and a broad downregulation of ULBP ligands for NK cells. These cells evade NK-cell-mediated clearance to remain latent for extended periods[50]. In this study, we also observed an association between WNT dysregulation and a stemness slow-cycling state and identified an unreported immune evasion of ESCC stem-like cells: masking the tumour cells with expression of NK cell markers. These findings, together with those previously discussed, highlight the critical interaction between key signalling pathways crucial for stem-cell propagation and the mechanisms that guide immune evasion. A deeper understanding of the unique interactions between cancer stem-like cells and the immune system could provide ground for developing therapeutic strategies that can harness the immune system against the "hardest immune evaders".

When investigating the differences in genomic changes among ESCC subtypes, we noted a strong positive association between *EP300* mutations/overexpression and stemness/NK marker XCL1-related signatures. Histone acetyltransferase p300 is a crucial transcriptional coactivator. Along with CBP, they regulate the transcription of thousands of genes in a cell via chromatin remodelling and histone modification, playing an important role in several fundamental biological processes, including proliferation, cell cycle, cell differentiation, and DNA damage response[51]. Of note, how cancer cells utilise p300 seems to be context-dependent; both loss- and gain-of-function genetic alterations in p300 have been reported across solid tumours and haematological malignancies[51–54]. Our data suggests that missense and splice site mutations of *EP300* appeared to have elevated expression levels of the mutated allele in RNA while truncating mutations of *EP300* had significantly decreased levels of the alternative allele in RNA. However, the overall mRNA expression of *EP300* was significantly higher in mutated than in wild-type samples on average. The *EP300* alterations potentially promoted ESCC tumours to become more stem-like phenotypes, which leads to immune exclusion, drug resistance and worse clinical outcomes. It has been shown that overexpression of EP300 led to upregulation of mesenchymal and stemness markers, increases in migration, invasion, anchorage-independent growth and drug resistance in a breast cancer cell model[55]. Similar observation was also obtained in non-small cell lung cancer and nasopharyngeal carcinoma cells[56,57]. EP300 knockdown, on the other hand, reduces cancer stem cell phenotype, EMT, tumour growth and metastasis in these

cancers[56–58], further supporting its oncogenic role and potential involvement in stem-like phenotype. Indeed, EP300 mutations and overexpression correlate with adverse prognoses in many solid cancers, including ESCC[33,34]. Importantly, it has been shown that pharmacological inhibition of CEP/p300 KAT activities sensitises cells to DNA-damaging chemotherapeutic agents and radiation[34,59]. Thus, more work lies ahead to test similar combinational therapies to treat these resistant stemness/*XCL1*-high ESCC cells in vitro and in vivo.

We also developed an amalgamated deep-learning model to extract and quantify significant histopathological features associated with each molecular subtype. This approach identified subtype-specific imaging features, and such features were highly enriched in their corresponding molecular subtypes. This AI model based on whole slide images highlighted the potential of predicting molecular subtypes using histological features only. Moreover, it also identified intratumoural heterogeneity in the tissue, as subtype-specific features seemed to be present in all slides but with varied proportions. Our ongoing effort is focused on analysing high-resolution image representations across all slides using hierarchical self-supervised learning[60] and testing if any of these features or combinations of features are correlated with clinical outcomes and molecular subtypes. Furthermore, a joint omics-image model based on multimodal deep learning[61] is yet to be developed using our data to allow for the most comprehensive data integration and biomarker discovery in ESCC.

In summary, this in-depth analysis of genomics-transcriptomics of Chinese ESCC patients provides insights into the nature of ESCC tumours. These findings pave the way for us to develop more effective diagnostic and therapeutic approaches for ESCC. In addition, the data created from this study, especially from treatment-naïve ESCC patients with follow-up more than four years after surgery, provides a unique public resource to better understand and treat ESCC.

## Methods
### Patient cohort
120 patients diagnosed with oesophageal squamous cell carcinoma from 2013 to 2016 were enrolled in Anyang Cancer Hospital under the approval of the ethics committee of Both Anyang Cancer Hospital and The First Affiliated Hospital of Zhengzhou University. None of these patients received any radiotherapy or chemotherapy before surgery, and pathology diagnosis was confirmed by three independent pathologists. Tumour samples and adjacent normal tissues at least 5 cm away from paired tumour tissues were collected and placed in liquid nitrogen within 30 minutes after the surgery operation. A validation cohort of 65 ESCC patient samples was also identified from Anyang Cancer Hospital, all primary treatment naïve tumours. All patients were informed and signed a patient informed consent, and the study was approved by the Ethics Committee of Zhengzhou University so that sex and age when first diagnosed were reported for each patient in Supplementary Data 1. However, no sex or age-specific analysis was carried out, as these were not associated with our molecular signatures.

### RNA-seq experiment
Total RNA from the tumour and matched normal samples were extracted with Invitrogen's TRIzol Regents according to the manufacturer's instructions. After quantification with Agilent 2100 Bioanalyzer (Agilent RNA 6000 Nano Kit), 1 ug RNA was used to construct the sequencing library following the introductions of VAHTS® Total RNA-seq (H/M/R) Library Prep Kit for Illumina® and quantified libraries were sequenced on Illumina X Ten platform (BGI) with paired-end 150 bp read length, with on average 120 M reads per sample.

### RNA-seq data analysis
Raw sequencing reads were first evaluated by FastQC(0.11.7)[62], and only clean data generated by SOAPnuke1.5.6[63] (-l 10 -q 0.5 -n 0.05 -Q 2 -G)

was aligned and quantified against the indexed GRCh37 genome using Salmon(version 0.9.0)[64]. The transcriptome abundances were imported into R (version 3.5.1) with 'tximport' R package[65]. Only transcripts with TPM larger than 1 in more than half of the samples were kept and normalised by cqn[66]. Principle component analysis (PCA) was carried out on the whole transcriptome to further explore the data quality and possible bias. Limma[67] method was then used to perform the different expression analyses for tumour versus normal pairs. Gene Set Enrichment Analysis (GSEA)[68] was further performed to identify significantly dysregulated canonical pathways.

### ESCC subtypes based on transcriptomics

To make sure that only the highly variable ESCC-related genes were used for this analysis, mean absolute deviation (MAD) for each gene among tumour samples was first calculated, and the top 1,500 variable genes with the largest MAD values were selected for the ESCC subtype discovery. The non-negative matrix factorisation (NMF) algorithm 'NMFConsensus'[69] was used to discover distinct transcriptomic subtypes with default parameters with cluster size $k = 2$ to 7 considered. Although $k = 2$ gave the best cophenetic correlation $r = 0.99$, $k = 4$ also achieved great clustering performance with cophenetic correlation $r = 0.985$ (Supplementary Fig. 1a), and the latter ($k = 4$) uncovered the level of heterogeneity and granularity of transcriptomic patterns in a finer resolution, thus was selected for our investigation. The rationalisation of these clusters was also verified by the consensus clustering method with Consenseclusterplus[70].

### Identification and annotation of subtype-specific genes

Limma was used to perform different expression analyses for each NMF cluster against all other clusters. Genes with adjusted $P$-value < 0.05 and log$_2$ fold change (FC) > 1 were considered as subtype-specific genes for that cluster. For the 'stemness' cluster, the cutoff of adjusted $P$-value < 0.05 and log$_2$FC > 0.8 was used to increase the number of representative genes for the cluster annotation. Two independent methods were applied for the functional annotation of clusters. First, for each cluster, representative genes were functionally annotated using DAVID[71]. Gene sets from Gene Ontology biological process terms, and KEGG pathways were used for the enrichment test. Second, the canonical pathways and cancer hallmark gene sets from the mSigDB database (v.6.2) were selected, and the over-representation hypergeometric test was applied to each subtype-specific gene set. The $P$ values from the hypergeometric tests were adjusted for multiple comparison testing using the Benjamini & Hochberg correction, and significant associations were reported at adjusted $P < 0.05$. The $P$ values were transformed as '−log$_{10}(P)$' and used for the heatmap. According to the annotation results, each NMF cluster was assigned a name representing their subtype transcriptomic features.

### Estimation of tumour infiltrating immune cell types and abundance

Previously defined immune cell type gene signatures[20–25,72] were used to deconvolute the immune microenvironment and estimate the immune cell abundance of all tumours with RNA-seq normalised gene expression data. For methods where the web service or R code was available, the default setting was applied for analysis. For the remaining published gene signatures, the mean of normalised expression of marker genes was calculated and used to correspond to the level of the represented cell types. Following a recent study[73], we further benchmarked the estimates/signatures of immune cell types from those methods using the following analyses. First, the signatures of various immune cells were correlated with tumour purity, with the aim to identify negative corelations between them for all immune cell types considered. Second, the immune signatures were correlated with the tumour copy number at the marker gene locus. A non-significant correlation was expected between them, to exclude any confounding

factors from tumour cells. Third, the immune estimates from all selected methods were correlated and compared, to measure the consistency among those methods for estimated cell types. Finally, the immune estimates were compared with the tumour-infiltrating lymphocyte (TIL) histopathology estimates of CD8$^+$ and CD4$^+$ T cells. The immune estimation that characterised our immune microenvironment the best (i.e., Danaher et al. signature) was chosen for all following analyses, covering 12 different immune cell types. In addition, as suggested by Rosenthal et al. and our benchmarking results, CD4$^+$ T cell estimates from Davoli et al.[24] were also included for our final immune profiling.

### ESCC subtypes based on immune cell estimation

Based on the immune estimates of 13 immune cell types across all our RNA-seq samples, consensus clustering using 'Consenseclusterplus'[70] was performed to identify distinct immune profiles among patients. The parameters of agglomerative hierarchical clustering, Pearson correlation distance and 50 re-samplings were applied. The size of the best-performing cluster was subsequently determined by consensus matrices and tracking plots. Immune cells were clustered with Pearson correlation and the 'average' clustering method. The rationality of our immune clustering was also verified by MCP-counter[23] derived immune cell estimates.

### Whole exome sequencing (WES) experiment

Genomic DNA from tumours and matched normal samples or peripheral blood were isolated using the QIAamp DNA Mini Kit (Qiagen), according to the manufacturer's instructions. To construct whole-exome capture libraries, 1 μg of genomic DNA from each fresh-frozen tumour and matched normal sample was randomly fragmented by Covaris into 250–300 bp. After fragmentation, Fragments were purified with the AxyPrep Mag PCR clean up Kit and then captured with the Agilent SureSelect Human Exomes V6 kit(~35.7 Mb, Cat No.: 5190-8881). All the constructed libraries were loaded on the Illumina X Ten platform (BGI Wuhan), and the sequences were generated as 150 bp paired-end reads.

### Whole-exome sequencing data analysis

After quality control with FastqQC (0.11.7), sequencing reads were mapped to the hg19 genome sequences with BWA (0.7.17)[74] mem followed by the further improvement of the alignments using GATK4 (4.0.6.0)[75] following its best practice guideline. For the identification of somatic variants, mutect2[76] and strelka2 (2.8.4)[77] were used, and only variants called by both variant callers were considered. Variant annotation was further performed using The Ensembl Variant Effect Predictor[78]. Three independent methods, MutSigCV[27,28], OncodriveFM[30] and dNdScv[29] were then used for the identification of driver genes (i.e., significantly mutated genes or functionally significant genes) in our WES cohort. For the analysis of copy number aberrations, Sequenza (2.1.2)[79] was used with default parameters. The tumour cellularity of all DNA samples was also derived from this analysis. Genome-wide gain or loss was defined as described previously[80]. Briefly, processed copy number values for each sample were divided by the sample mean ploidy and log$_2$ transformed. Gain and loss were defined as >log$_2(2.5/2)$ and <log$_2(1.5/2)$, respectively. Amplification was defined as ≥log$_2(4/2)$ and deletion as ≤log$_2(1/2)$. Clonality analysis for each tumour was further carried out using PyClone[81]. Clusters with fewer than 3 mutations were filtered out. Major clones and subclones were identified based on the cluster cellular prevalence derived from PyClone.

### Pathway functional mutation enrichment analysis

We chose the 50 cancer hallmark gene sets curated at the mSigDB database for this analysis. Using the called mutations, we considered all non-silent mutations as functionally relevant. For each sample, a

functional mutation enrichment score for each gene set was derived by dividing the number of functional mutations by the number of genes in the gene set, further adjusting for the total mutation burden for that sample. For each gene set, the functional mutation enrichment scores were then compared among the four subtypes using the Kruskal−Wallis test, to identify whether any gene sets were specifically enriched for certain subtypes. The correlation between function mutation enrichment score and tumour cellularity was calculated across all patients for each gene set. The GSVA[82] R package was used to calculate pathway activity scores of the 50 gene sets across all samples, and GSVA scores were subsequently compared among the four subtypes for each gene set using the Kruskal−Wallis test.

### Estimation of intra-tumour heterogeneity

Shannon diversity was used to calculate the intra-tumour heterogeneity using a previously described method[83]. Briefly, the variant allele frequencies of all mutations in each tumour sample were assigned into 10 bins of equal range, starting from 0-10%, 10-20%, to 90-100%. Diversity index H′ was calculated based on the proportion of mutations in each bin ($p_i$) with the formula: $H' = -\sum_{i=1}^{n} p_i \ln p_i$.

### Data analysis of the CCLE dataset for *XCL1* expression

Gene expression profiling (in the RPKM unit) of the CCLE dataset was downloaded from its data portal (https://portals.broadinstitute.org/ccle), and expression values were transformed as $\log_2(RPKM + 1)$. Expression for 22 oesophageal SCC cancer cell lines was extracted, and *XCL1* expression was inspected. The 22 cell lines were further split into *XCL1* high ($n = 11$) and low ($n = 11$) groups based on $\log_2(RPKM + 1)$ value of 1.5 as cutoff, and differential expression analysis in the whole transcriptome was performed using limma. GSEA analysis was further carried out for gene sets from canonical pathways curated in the mSigDB database.

### Deep learning histology analysis

**Image pre-processing.** Stained (H&E) and scanned whole slide images (WSI) of FFPE tissue slides were obtained in SVS format for our samples. Slide layers corresponding to 64× down sample factor of 20× objective power and 0.44 μm per pixel resolution were extracted using the Openslide python package[84]. All images were down-sampled to 64× factor and converted to JPEG format for easy manipulation and handling. Each WSI was manually reviewed by a qualified pathologist, and poor-quality images were discarded under the direct pathologist's supervision. The poor quality of imaging means the sections were folded without clear morphology, or there were not enough tumour cells presented in the slides obtained from the department of histopathology. Only those images with tumours and free of technical artefacts were used for further analysis. A total of 91 WSIs were retained for the deep-learning analysis, i.e., differentiated group, $n = 28$; immunogenic, $n = 27$; metabolic, $n = 18$; stemness, $n = 18$. The slides were tilled in non-overlapping 300 × 300 pixels windows and filtered for information content, i.e., we removed all the tiles with >20% background or irregular tissue coverage. The number of tiles depends on the area covered by tissue and can vary from tens to hundreds. Thus, we randomly selected 50 tiles per WSI, considering the required memory and the average number of tiles per slide (total = 4550 tiles in 91 patients). We used 65% of those tiles for discovery and 35% for testing. The data split was performed at the patient level to prevent overlaps between the two sets.

**Feature extraction.** We performed feature extraction using five state-of-the-art convolutional networks, namely Inception-V3, Inception-ResNet-V2, DenseNet-121, VGG16 and ResNet-50. Already pre-trained for any image analysis, combining these networks allow us to obtain 7169 features of a wide range of characteristics. Thus, we obtained 50 (tiles) × 7169 (features) for each slide, ready for inference of our gene expression classification.

**Inference of gene expression classification.** We used the Wilcoxon test to identify features associated with the gene expression classification/subtypes. During the inference on the discovery set, we create a binary variable (target group = 1 and the rest = 0) for each group (DIFF, IMM, STEM and MET) and features (7169). A *P*-value was calculated by evaluating the difference in the average feature level between the group and the rest. The resulting P values were then adjusted to control for the FDR across the entirety of feature-group pairs tested using the method of Benjamini and Hochberg. We selected the top five features per group that are significant at FDR < 0.01, higher in the target group than the rest, and unique to one group (Supplementary Data 4). We summed these five features to create four stable histological markers of gene expression-based classifications (Meta Features). Finally, we repeated the above analysis on the left-out test set and compared the direction and significance of these features to the discovery set (Supplementary Data 5).

### Single-cell RNA-seq analysis

Processed unique molecular identifier (UMI) matrices and associated cell annotation were downloaded from Gene Expression Omnibus under the accession of GSE160269. Processed data were further loaded into Seurat R package[85] for downstream analysis and visualisation. UMAP was used to visualise all annotated cellular clusters and feature plots of marker genes were subsequently generated. For the gene set scoring, the Seurat AddModuleScore function was used, and the gene sets of interest, e.g., cell cycle activities, were available from the mSigDB database. *XCL1* positive and negative cells were identified based on the Seurat SCTransform-based normalised gene-level counts: as positive when normalised gene-level counts >0; and negative when normalised gene-level counts = 0. Wilcoxon rank test was then used to compare the level of cell cycle activities, i.e., gene set scoring, between the two groups.

To investigate the heterogeneity of transcriptional programmes of ESCC epithelial cells, the NMF clustering with $k = 10$ factors was performed on epithelial single-cell data, similar to previous investigations[86–88], followed by the differential expression analysis using the Seurat "FindMarkers" function to identify top differentially expressed genes, as well as gene set enrichment analysis for each cluster. Signature genes for each cluster were identified based on adjusted *P*-value < 0.0001 and $\log_2$ fold change >1. The top 50 signature genes were then selected based on the $\log_2$ fold change for each cluster. The NMF clusters were annotated based on their signature genes and up/down-regulated pathways.

### Drug sensitivity analysis

To evaluate the cytotoxicity of chemotherapy drugs in human ESCC cell lines, ESCC cancer cells KYSE-150, KYSE-180, KYSE-410 and their corresponding XCL1 overexpressed cells were seeded in 96-well plates at 4000 cells/well, and cultured in DMEM with 10% FBS for 24 h, then treated with various concentrations of 5-FU for 72 h in a 37 °C incubator with 5% $CO_2$. Cell viability was examined using the MTS assay (Promega, Madison, WI, USA). The IC50 value (half maximal inhibitory concentration) was calculated. Experiments were performed in triplicate. Cell viability in each well was calculated based on the following formula: Cell viability = (absorbance value of treated cells – background)/ (absorbance value of untreated control cells – background) and expressed as a percentage of that for untreated cells. The differences in cell viability between the control cells and the corresponding XCL1 overexpressed cells were determined using the Mann−Whitney test.

To explore the sensitivity of ESCC cells against different drugs, the drug screening data of IC50s was downloaded from the Genomics of Drug Sensitivity in Cancer (GDSC) data repository

(https://www.cancerrxgene.org/downloads/bulk_download). The GDSC1 data set was used. The *z*-transformed IC50 values were compared between *XCL1* high and low samples using the Student's *t*-test.

### Establishment of XCL1 overexpressing stable cell lines and its expression detected by Quantitative real-time Polymerase Chain Reaction (qRT-PCR)

Human XCL1 cDNA was synthesised by GENEWIZ and cloned into a lentivirus vector. To obtain XCL1 overexpressing or control ESCC stable cell lines, $1 \times 10^5$ cells were seeded into 24-well plates and infected with lentivirus at MOI = 50 (Multiplicity of infection). 72 hours later, cells were selected by puromycin (SelleckChem) and relative expressing of XCL1 was elevated by qRT-PCR assay. In brief, cell total RNA was isolated by TRIzol™ Reagent (ThermoFisher Scientific) and reverse transcriptase into cDNA using PrimeScript™ RT reagent Kit (Takara). Then XCL1 level was further quantified by TB Green® Premix Ex Taq™ II kit (Takara) with $2^{-\triangle\triangle CT}$ assay following the instructions.

### Western blot assay

Total cell protein was isolated with RIPA lysis buffer (Beyotime Biotechnology, P0013C) containing 1% protease inhibitor cocktail solution (Roche, 04693132001). Then 30 μg protein prepared with loading buffer was separated by 10% SDS-PAGE and transferred to PVDF membranes (Millipore). After blocking with 5% non-fat milk, membranes were incubated with primary antibodies at 4°C overnight, washed and probed with horseradish peroxidase-conjugated secondary antibodies (1:5000, ZSBIO), then detected by the enhanced chemiluminescence (ECL) system (Thermo pierce, USA). Antibodies were listed as the following, anti-SFRP1 (1:500, Atlas antibodies, HPA064870), anti-GAPDH (1:5000, ProteinTech, 60004-1-Ig), HRP-Goat Anti-Rabbit IgG (1:5000, ZSBIO, ZB-5301), HRP-Goat anti-mouse IgG (1:5000, ZSBIO, ZB-5305).

### Cell proliferation assay

To validate the effect of SFRP1 or XCL1 on cell proliferation, cells were seeded into 96-wells plate at $5 \times 10^3$/well, and the real-time monitoring of cell proliferation was performed with Incucyte® Live-Cell Analysis Systems. All experiments were performed three times in triplicates.

### Edu Incorporation and cell cycle assays

We detected the proliferation of KYSE180, KYSE180-XCL1, KYSE410 and KYSE410-XCL1 cells in the BeyoClick Edu cell proliferation detection kit (Beyotime Biotechnology, C0075S) according to the instructions. Briefly, we incubated the cells cultured in 6-well plates with 10 μM Edu working solution for 2 h and then did immunofluorescence. We got about 5 visual fields of each cell using a fluorescence microscope and calculated the Edu-positive cells in each field using ImageJ 1.53e.

The cell cycle assay was detected by Propidium iodide(PI) staining using the Annexin V-FITC/PI Apoptosis Detection Kit (A211-01, Vazyme Co., Nanjing, China) according to the instruction. Then we detected the cell percentage of each phase by flow cytometry and analysed by Flowjo 10.53 software.

### Xenograft tumour growth of ESCC in nude mice

SFRP1 cDNA and shRNA sequence were synthesised by GENEWIZ and cloned into lentivirus vector to obtain SFRP1 overexpressing (KYSE-70 and KYSE-140) and knockdown (KYSE-450 and KYSE-520) cells. Five-week-old female BALB/c Nude mice were purchased from Beijing Vital River Laboratory (Beijing, Cat# 401). Sex analysis was not considered. $2 \times 10^6$ of SFRP1 overexpressing/knockdown and its matched control ESCC cells were subcutaneously injected into bilateral flanks of BALB/c nude mice ($n = 6$ per group). Tumour volumes were measured using electronic callipers [tumour volume = (length x width$^2$xπ)/6] twice a week. When tumours reached 3000 mm$^3$, tumour ulceration occurred, or animals lost 20% of their body weight, animals would be sacrificed.

In this study, mice were sacrificed, and xenograft tumours were photographed 30 days after the tumour cells were transplanted subcutaneously. All mice in the overexpressing group developed tumours, while two mice in the knockdown group had no tumour formation. The animal study was approved by the Animal Welfare and Research Ethics Committee of Zhengzhou University (Zhengzhou, China) for the project SQ2016YFHZ020118 and was conducted in accordance with accordance with the Guide for the Care and Use of Laboratory Animals.

Sequence for SFRP1 shRNA were as following, SFRP1-shRNA1#: CCGGCCCTGTGACAACGAGTTGAAACTCGAGTTTCAACTCGTTGTCA-CAGGGTTTTTG, SFRP1-shRNA2#: CCGGCCGGAGAGTTATCCTGA-TAAACTCGAGTTTATCAGGATAACTCTCCGGTTTTTTG.

### Immunohistochemical assay

Hematoxyline and Eosin (H&E) and Immunohistochemistry staining were performed as standard protocol. Briefly, ESCC patients' sections were deparaffinised, rehydrated, heated for antigen retrieval and blocked endogenous peroxidases. After blocking with normal goat serum, slides were incubated overnight at 4 °C with proper primary antibodies. Subsequently, slices were stained with ElivisionTM plus (KIT-5020, Maixin Biotechnology), DAB kit (DAB-1031, Maixin Biotechnology) and sealed with neutral resins, then photographed with NDP.view2 system. Some slides for XCL1 and LRG6 staining were scanned by NanoZoomer S210 (Hamamatsu Photonics), which is an automated bright field slide scanner and then analysed by QuPath and NanoZoomer Digital Pathology Image (.ndpi) software. Antibodies used in this study were listed as follows, anti-SFRP1 (1:200, Atlas antibodies, HPA064870), anti-XCL1 (1:400, Atlas antibodies, HPA057725), anti-LGR6 (1:100, Abcam,126747), anti-CD160 (1:300, Origene, TA349762), anti-CD8 (Genetech, GT211207), anti-CD4 (Maixin Biotechnology). For pathology, TIL estimates, positive cells in 15 random fields were counted with Image J software.

### TCGA data

RNA-seq data were downloaded from the UCSC Xena Browser, TCGA Oesophageal Cancer (ESCA) ($n = 198$). The available 'Level_3' normalised gene-level data were obtained, and the 90 annotated oesophageal SCC samples were further extracted based on the clinical information from the TCGA integrated genomic study[13].

### Survival analysis

The Kaplan–Meier method was used to generate curves for overall survival with R survminer 0.4.3 package. Surv_cutpoint function was used to split each endpoint with numeric values into low and high groups. Hazard-ratio (HR), 95% confidence interval and log rank *p*-value for each endpoint studied were calculated with Cox proportional hazard (PH) regression mode.

### Statistics and reproducibility

All statistical analyses were performed in the R programming environment (https://www.r-project.org/). No statistical method was used to predetermine sample size. The experiments were not randomised.

### Reporting summary

Further information on research design is available in the Nature Portfolio Reporting Summary linked to this article.

## Data availability

Sequence data used during the study has been deposited at the National Genomics Data Centre of China (https://bigd.big.ac.cn/), and the Bioproject Access ID is PRJCA001577. This BioProject has two associated Genome Sequence Archive (GSA) accession numbers: HRA000111 hosts the raw sequencing data of RNA-seq, while HRA000112 has the raw sequencing data of whole exome sequencing. The availability of the data has been approved by the Human Genetic

Resources Registration System of the Ministry of Science and Technology of the People's Republic of China with the registration number 2024BAT00864. Single-cell RNA-seq data were downloaded from Gene Expression Omnibus (GEO) under the accession number GSE160269. The data associated with PRJCA001577, i.e., GSA accession numbers HRA000111 and HRA000112, was under controlled access. To access this data, interested users must apply through the National Genomics Data Centre of China GSA system. The application process involves providing a detailed research proposal that outlines the purpose and intended use of the data. The Data Access Committee (DAC), with the identifier HDAC000064, reviews these applications to ensure that the proposed research aligns with the ethical and scientific standards set for the use of such data. Data access will be updated on a yearly basis with renewed options at the end of each year. Three publicly available gene expression data sets from GEO were used: GSE53625, GSE47404, and GSE160269. TCGA Oesophageal Cancer (ESCA) RNA-seq data was also used [https://xenabrowser.net/datapages/?dataset=TCGA.ESCA.sampleMap%2FHiSeq&host=https%3A%2F%2Ftcga.xenahubs.net&removeHub=https%3A%2F%2Fxena.treehouse.gi.ucsc.edu%3A443], and 90 ESCC samples were further extracted. Source data are provided with this paper.

## Code availability

All the codes that were used for analyses and the generation of figures are available at https://github.com/Zhong2020/ESCCproject. All the codes used for the deep-learning analysis are available at https://github.com/BioInforCore-BCI/giExtract. (https://doi.org/10.5281/zenodo.11049708)[89].

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

## Acknowledgements

This project is supported by the National Key R&D programme of China (2016YFE0200800), the Nature Sciences Foundation of China (U1704282, U1904148, 82172941 and 81771776) and the core funding for the development of the Cell and Gene Therapy Programme by Zhengzhou University. JW, NRL and YW also acknowledge support from the Cancer Research UK City of London Centre of Excellence Award core funding to Barts Cancer Institute. JW acknowledges support from the Academy of Medical Sciences Springboard Award (SBF003\1025), and Cancer Research UK (C355/A26819) and FC AECC and AIRC under the Accelerator Award Programme. LSCD and YW acknowledge the support from the MRC (MR/V006053/1), ZW acknowledges the Henan Provincial and Ministry co-constructed youth projects for medical science and technology (SBGJ202103031).

## Author contributions

Y.W. and J.W. conceived and supervised this study. Y.W., J.W., and W.L. were responsible for all aspects of study design and managed the project with the support by D.G.; G.J. and W.W. were in charge of the collection of all samples and clinical data, DNA and RNA extraction, with the help from P.C., Z. Zhang and J.L. Z. Wang led the bioinformatic analysis with support by Y.D. and X.X. under the supervision by J.W.; F.K. and E.B. performed the single cell RNA-seq analysis, C.A.A. and Y.C. developed the deep-learning model analysing whole slide images under the supervision by J.W. and Y.W. Z.C. led and performed the in vitro and in vivo biological function validation studies with P.W., H.X., Z. Zhang, S.L., J.M., Z. Wang and P.L.; D.G. and R.G. did histopathology and IHC staining, Y.W., W.L., Z.C., R.G. and Y.W. reviewed H&E and IHC slides and prepared the representative pictures; L.Z., L.S.C.D., Y.G., J.D. and N.L. participated in interpretation of some experiments and critically reviewed the paper. J.W., Z. Wang, Z.C. and Y.W. interpreted all results and wrote the paper.

## Competing interests

The authors declare no competing interests.

## Additional information

¹Department of Pathology, The First Affiliated Hospital of Zhengzhou University, Zhengzhou 450052, People's Republic of China. ²Department of Molecular Pathology, The Affiliated Cancer Hospital of Zhengzhou University & Henan Cancer Hospital, Zhengzhou 450008, People's Republic of China. ³National Centre for International Research in Cell and Gene Therapy, School of Basic Medical Sciences, Zhengzhou University, Zhengzhou 450052, People's Republic of China. ⁴Sino-British Research Centre for Molecular Oncology, Academy of Medical Sciences, Zhengzhou University, Zhengzhou 450052, People's Republic of China. ⁵State Key Laboratory of Esophageal Cancer Prevention & Treatment, Academy of Medical Sciences, Zhengzhou University, Zhengzhou 450052, People's Republic of China. ⁶Centre for Cancer Genomics and Computational Biology, Barts Cancer Institute, Queen Mary University of London, London EC1M 6BQ, United Kingdom. ⁷Centre for Biomedical Science Research, Leeds Beckett University, Leeds LS1 3HE, UK. ⁸CAS Key Laboratory of Infection and Immunity, Institute of Biophysics, Chinese Academy of Sciences, Beijing, People's Republic of China. ⁹Department of Pharmacology, School of Basic Medical Sciences, Academy of Medical Sciences, Zhengzhou University, Zhengzhou 450001, People's Republic of China. ¹⁰Centre for Cancer Biomarkers & Biotherapeutics, Barts Cancer Institute, Queen Mary University of London, London EC1M 6BQ, UK. ¹¹Department of Molecular Pathology, Anyang Cancer Hospital, Anyang City 455000 Henan Province, People's Republic of China. ¹²Department of Cardiology, Centre for Cardiovascular Diseases, The First Affiliated Hospital of Zhengzhou University, Zhengzhou 450052, People's Republic of China. ¹³Henan Key Laboratory of Hereditary Cardiovascular Diseases, The First Affiliated Hospital of Zhengzhou University, Zhengzhou 450052, People's Republic of China. ¹⁴Department of Cardiology, Beijing Anzhen Hospital, Capital Medical University, No. 2, Anzhen Road, Chao Yang District, Beijing 100029, People's Republic of China. ¹⁵These authors contributed equally: Guozhong Jiang, Zhizhong Wang, Zhenguo Cheng, Weiwei Wang. ¹⁶These authors jointly supervised this work: Wencai Li, Jun Wang, Yaohe Wang. ✉e-mail: liwencai@zzu.edu.cn; j.a.wang@qmul.ac.uk; Yaohe.wang@qmul.ac.uk

