## [Peer Review File · Nature Communications]

Editorial Note: Figure on page 39 of this Peer Review File have been redacted as indicated to remove third-party material where no permission to publish could be obtained.

"Chen, D., et al. An Ensemble Deep Neural Network for Footprint Image Retrieval Based on Transfer Learning. Journal of Sensors 2021, 6631029 (2021)."

REVIEWER COMMENTS

Reviewer #1 (Remarks to the Author): Expert in oesophageal cancer genomics

Jiang et al. performed whole-exome sequencing and transcriptome analysis on Chinese 120 ESCC samples to characterize ESCC in detail. The authors classified four subtypes based on the expression and three based on the estimated immune cells in cancer tissues. They found that the stemness subtype showed poor prognosis and enriched in C3 immune subtype. In addition, they showed some relationships between transcripts and subtypes and analyzed them functionally. Moreover, they found NK cell-like expression signature in cancer cells. Genetic analysis also gave some insights. However, overall, their claims are not well supported by the results, and there are many overstatements as described below.

Major points:

Line 97-98 The authors classified ESCC based on 1500 gene expressions with the highest MAD score and adopted four subtypes. The methods section describes that the cluster number with the highest cophenetic coefficient was adopted. As shown in Extended Data Figure 1a, the cophenetic coefficient for $k=2$ is 0.9928 and $k=4$ is 0.985, but they adopted $k=4$ instead of $k=2$. Is there any other reason why the authors adopted $k=4$?

Line 110-111 Fig. 1b does not show "interferon gamma pathway" or "chemokine signaling pathway". Were these pathways significantly downregulated in the stemness subgroup?

Line 115-120 The authors claimed histological characteristics of each subtype. However, there is little evidence to support their claim because they just showed the representative images and did not show any quantitative data extracted from histological images.

Line 124-127 The result is unreliable because functional analysis of SFPR1 overexpression was performed in just one cell line. Also, it is highly recommended to show the result of the knockdown experiment using more than one effective shRNA to exclude off-target effects.

Line 154 The association between "stemness" and "NK cell estimates" is not strong because the correlation r -value of 0.32 – 0.4 is not high.

Line 165-166 The authors showed immunohistochemical images of Sample 401 in Fig.2g and Extended Data Fig. 5c. How is the difference of LGR6-staining in the same tumor explained? Also, the architecture of cancer cells is different. Are these derived from the same tumor sample?

Line 198-201 The result is incredible because the authors performed overexpression of XCL1 in just one cell line. In addition, it is strongly recommended to show the result of the knockdown experiment of XCL1 to support their claim.

Line 247-250 Among EP300 mutations, 30 % were truncating mutations or deletion. The authors showed higher expression of the mutated allele. However, if the cancer had a truncating/deletion mutation, the

mutated allele is expected to result in loss of function. Therefore, their claim of a gain-of-function mode of EP300 mutations is not supported.

Line 254-277 The authors showed a higher frequency of NOTCH1 mutation as well as EP300 mutation in the stemness subgroup. Therefore, these results may derive from the effect of NOTCH1 mutation. To investigate the effect of EP300 mutation, it is reasonable to compare EP300-wild and mutant cancers among the same subgroup because the genetic backgrounds of subgroups are different from each other.

Line 296-299 In Fig. 9c left panel, three subtypes other than the stemness subtype showed a comparable frequency of truncating mutations in wnt pathway genes. Therefore, because the prediction of the effect of mutations is more reliable for truncating mutation than for missense mutation, it seems that deleterious Wnt pathway mutations are similar.

Minor points:

Line 99-101 They designated four subgroups based on known functional gene expression rather than GSEA.

Reviewer #2 (Remarks to the Author): Expert in tumour immunology, NK cells, and oesophageal cancer

Comments for the manuscript entitled “Stemness and NK-like signatures define the poorest subtype of oesophageal squamous cell carcinoma” by Guozhong Jian et al.

This referee aims to comment on the non-bioinformatical part of this manuscript, leaving the comments for the precise bioinformatics to experts on bioinformatics and statistics.

The manuscript presents an impressive molecular investigation of oesophageal squamous cell carcinoma (ESCC) from 120 Chinese patients. The authors have performed transcriptomics with GSEA and immune profiling, and mutational enrichment analyses, with the aim to find subtypes of tumors, and link this information to novel immune evasion strategies or drug sensitivity. The focus of the study is using bioinformatics, and the results from the authors own patient cohort is compared to previous studies on related cohorts, especially regarding the immune profiling strategy. There is no doubt that the study is of large importance, with ESCCs being one of the deadliest cancers worldwide today, with no effective therapies at hand. A study of this size, at this detailed molecular level, is difficult to find in the literature and will make a difference. Whereas the bioinformatics part of this study seems strong, the immunology and pathology sections are less robust. The manuscript is written in a very condensed scientific English, which makes it difficult to actually understand what has been done in the various panels of experiments. This also makes the reader insecure of what the authors actually have accomplished (the materials and methods (and figure legends) does not help or guide). To be able to fully understand this study, the manuscript has to be rewritten in a more detailed manner. Some comments are given below.

1) Every panel of each figure needs to be explained better, both experimentally and statistically, to be able to understand what has been done. This goes for the material and methods part too. Controls should be added where appropriate, and in cases where a value is calculated compared to a control, the

control needs to be described equally well. Full statistics should be explained also in Figure legends.

2) Row 114. "The molecular subtypes present distinguished histological features". Is this based on these four photographs of four tumors? How representative are they? Where are the statistics behind this to be able to state this quite important finding? The authors should have access to all tumors, and thus be able to perform a tumor tissue array of paraffinized samples, to compare the histology of all 120 tumors on one single glass slide using a light microscope. This way statistics can be added.

3) Row 116-120. The authors describe these histological features using nomenclature of keratin pearls, extensive immune infiltration etc, without showing it or even presenting how this was done or annotated. Again, to say this a tumor tissue array would be good and should be annotated and evaluated statistically. The material and methods section should be updated accordingly, presenting how keratin pearls or immune cell infiltration was assayed.

4) Row 126. Xenograft models are presented in the text, but no ethical permit or precise description of how it was exactly performed is present in the text or materials and methods. Just very vague descriptions referring to Extended data Fig 3. What is Ext data Fig 3a? Explain each panel. Extended data Fig 3 is a typical representative for the concerns mentioned above of too limited explanations.

5) Figure 2e, row 149, is this a Fisher's exact test? Where are the statistics shown in the figure? The figure legend is minimal.

6) Fig 2g. IHC controls are absent. A normal tissue should be used as control, and a cell line lacking (negative) or expressing (positive) should be shown to trust the IHC antibody specificity. Again, a tissue tumor array would be good to be able to conclude the results, only four samples are not enough to conclude.

7) Row 158-166. How is this IHC annotation performed and shown? The authors show statistics in the text that is not described in detail, and cannot be found in the figures or legends. Is it Pearson Chi-square tests or what do authors mean with co-expression? Tables of annotations? This needs to be clearly shown.

8) From row 177 and on "NK-like tumor cells" are used to describe tumor cells expressing CD160. It would be preferable to not name the tumor cells "NK-like tumor cells", since they are tumor cells and not lymphocytes.

9) Fig 3f. How is the IC50 experiment performed exactly. The materials and methods do not describe what is being shown in panel 3f. How are the numbers calculated? At what concentration, time and compared to what? If a Mann-Whitney test is used, what exact control are the bars compared to?

10) Figure 4 presents the genomic mutational landscape. This is important. The authors continue with XCL1 as a promising biomarker for drug sensitivity, using cell lines, and EP300 for survival in the patient cohort. How can these be linked in patient material and survival?

11) Figure 5, the authors present immune infiltration and association with clonal diversity, and is also important. It would be good to perform IHC panels on some basic immune populations of the tumors to be able to support the data from the Shannon diversity index. Immune exclusion of T cells compared to M2 macrophages, as well as NK cells would be interesting to see. If CD160 is expressed on tumor cells.

12) The manuscript needs to be rewritten, to add much more detail for each panel, to be able to understand what has been done. As it is now, it is difficult to interpret the results.

Reviewer #3 (Remarks to the Author): Expert in cancer stem cells

In this study Jiang et al., aims at defining novel molecular subgroups in human Oesophageal Squamous Cell Carcinoma (ESCC) by analysing the transcriptome and genome of a big cohort of ESCC from Chinese patients.

The authors, using bulk-RNA sequencing, identify four distinct ESCC transcriptomic subtypes. Among them one shows the worst survival rate and is characterized by the expression of WFDC2, SFRP1, LGR6 and VWA2 genes (“stemness” subtype). The data obtained from the RNA sequencing analysis is then analysed using deconvolution methods to determine the tumor immune composition of the ESCC samples. Using this approach, they divide the tumors in three categories according to the content/type of immune cells present within the sample and identify a group of tumors showing high expression of NK-like gene signature, the group associated with the poorest survival rate. Interestingly, they observe that tumors showing high expression of XCL1 (gene upregulated within the NK gene signature and C3 immune type) present higher resistance to several drugs. Thirdly, the authors characterize the genomic landscape of the same cohort of patients and describe genomic landscapes similar to those described in the numerous genomic analyses that have been reported on ESCC (The Cancer Genome Atlas Research, N. et al. *Nature* 541, 169–175 (2017); Song, Y. et al. *Nature* 509,91–95 (2014); Lin, D. C. et al. *Nat. Genet.* 46, 467–473 (2014); Gao, Y. B. et al. *Nat. Genet.* 46,1097–1102 (2014)). Finally, the authors determine if there is any correlation between the transcriptomic and immune signatures and the mutational landscape of the ESCC analysed.

The main criticism is that this study lacks accuracy as many of the statements are not well sustained or only based on correlations. There is a lack of validation of the transcriptomic and immune signatures in biological samples and lack of experiments in biological samples aimed at supporting the numerous correlations present in the study. These validations and experiments would certainly help to strengthen and confirm the message of the paper.

Remarkably, there are some points that need to be further developed or clarified:

1-Using bulk RNA-sequencing the authors classify the ESCC analysed in 4 different subtypes characterized by the expression of different markers, subtypes that show different morphological characteristics and patient survival rates (Figure 1). The authors claim that they are interested to characterize both the tumor and tumor microenvironment. However, they do not analyse if the genes found to be upregulated in the different subgroups are expressed by the tumor cells or by the tumor microenvironment, which is extremely relevant to validate and strengthen this key point the authors want to make. This validation could be accomplished by performing immunostainings for the different markers in tumors from the 4 ESCC subtypes or alternatively using techniques that would allow a deeper characterization of the different cell populations present in the samples (single cell RNA sequencing). Using the latter approach, a recent paper (Zhang, X., Peng, L., Luo, Y. et al. *Nat Commun* 12, 5291, 2021) identified eight different common expression programs of epithelial cells in ESCC tumors and characterized the populations present in the tumor microenvironment of the same tumors.

2- In Figure 2 the authors identify a cluster, cluster 3, characterized by the expression of NK markers,

again it is not clear if these NK markers are expressed by the tumor cells (staining for XCL1 and Lgr6 Fig2 g superior panel is too weak to be able to conclude anything and morphologically the tumor shown does not seem a “stemness” tumor) or if there is infiltration of NK cells or presence of NK cells surrounding the tumors (staining for a NK marker would be extremely helpful). Therefore the description of the immune infiltration and immune microenvironment is superficial and needs to be improved. In addition, only 50% of the stemness-subtype tumors could be classified as belonging to the immune cluster 3 (Fig 2e), so the statement “given the strong association between stemness and NK cell estimates- line 154-155” is overstated and should be corrected.

3- One of the genes that characterize Cluster 3 (NK-like signature) is XCL1. The authors show using GSEA that XCL1-high cells exhibit upregulation of drug metabolism of cytochrome P450, retinol metabolism and biological oxidations (line 185) and in line 106 the authors state “ the metabolic subtype is associated with the upregulation of genes involved in drug metabolism by cytochrome P450 and retinol metabolism”. Would this mean that tumors showing high XCL1 expression belong to the “metabolic subtype”? This is confusing and highlights that the link between XCL1 (NK-like) and “stemness subtype” or “metabolic subtype” should be better explained.

4- In Figure 2g, Extended Data Figure 5c and line 164-172, the authors state that XCL1 is exclusively expressed in cancer cells showing adenocarcinoma morphology. Should we consider all XCL1-expressing tumours adeno-squamous cell carcinomas? If so, the authors should repeat the analysis by studying the ESCC and adeno-squamous cell carcinomas separately as relevant differences may be found in terms of transcriptomics, genomics and response to chemotherapy.

5- The authors suggest that XCL1-high cells are slow cycling (line 191-193). This is an important point that could explain the higher chemotherapy resistance observed (Fig3). I would recommend the authors to study this point in more detail as it is quite relevant and could increase the impact of the study. BrdU/EdU incorporation studies can help to uncover this matter.

Point by point responses to the reviewers' comments

REVIEWER COMMENTS

Reviewer #1 (Remarks to the Author): Expert in oesophageal cancer genomics

Jiang et al. performed whole-exome sequencing and transcriptome analysis on Chinese 120 ESCC samples to characterize ESCC in detail. The authors classified four subtypes based on the expression and three based on the estimated immune cells in cancer tissues. They found that the stemness subtype showed poor prognosis and enriched in C3 immune subtype. In addition, they showed some relationships between transcripts and subtypes and analyzed them functionally. Moreover, they found NK cell-like expression signature in cancer cells. Genetic analysis also gave some insights. However, overall, their claims are not well supported by the results, and there are many overstatements as described below.

Major points:

1. Line 97-98 The authors classified ESCC based on 1500 gene expressions with the highest MAD score and adopted four subtypes. The methods section describes that the cluster number with the highest cophenetic coefficient was adopted. As shown in Extended Data Figure 1a, the cophenetic coefficient for $k=2$ is 0.9928 and $k=4$ is 0.985, but they adopted $k=4$ instead of $k=2$. Is there any other reason why the authors adopted $k=4$?

Response: We thank the reviewer for such detailed observation. Our choice of $k=4$ used in NMF was based on the combination factors of NMF cophenetic coefficient and the heterogeneous molecular signatures sufficiently revealed by our data. When $k=2$ was used, based on the signature gene sets of the two groups and their functional annotation, only two molecular subtypes were revealed, differentiated and immunogenic. The signatures of metabolic and stemness subtypes could not be identified by $k=2$. This is also supported by the Supplementary Table S1 of this rebuttal that the top signature genes of our differentiated and immunogenic subtypes were still significant genes for the two subgroups when $k=2$, but top signature genes of the metabolic and stemness subtypes were not significant anymore when $k=2$. Most importantly, our four transcriptomic subtypes and their signatures were successfully validated by three independent cohorts (Extended Data Figure 1b), and were strongly supported by the following clinical, histopathology, *in vitro* and *in vivo* (for stemness) experiments.

Supplementary Table S1. Comparison of signature genes of our four subtypes between k=4 and k=2 in NMF

Signature genes	log2FC (one group vs others)	P.Value	adj.P.Val	k=4 groups	k=2 groups	log2FC (group2 vs group1)	P.Value	adj.P.Val
LCE3D	4.292	1.54E-12	2.95E-10	Differentiated	Differentiated	-2.419	0.00016296	0.00252447
CDSN	3.966	7.01E-14	1.05E-10	Differentiated	Differentiated	-2.300	4.28E-05	0.00084861
KLK5	3.884	7.18E-13	1.79E-10	Differentiated	Differentiated	-2.316	4.81E-05	0.0009412
SPRR2G	3.685	1.02E-09	4.77E-08	Differentiated	Differentiated	-1.724	0.00628954	0.03822871
DSG1	3.590	2.00E-11	2.00E-09	Differentiated	Differentiated	-1.525	0.00717146	0.04182777
MS4A1	2.591	1.56E-12	4.04E-11	Immunogenic	Immunogenic	2.503	5.35E-14	1.09E-11
CD79A	2.558	1.16E-19	5.82E-17	Immunogenic	Immunogenic	2.352	2.05E-20	7.17E-17
CXCL9	2.263	8.70E-12	1.95E-10	Immunogenic	Immunogenic	2.117	2.20E-12	2.82E-10
MZB1	2.262	7.31E-22	1.10E-18	Immunogenic	Immunogenic	2.081	1.01E-23	2.81E-19
IDO1	2.206	1.48E-09	1.94E-08	Immunogenic	Immunogenic	2.444	3.15E-14	6.79E-12
GSTA1	3.956	3.37E-09	2.10E-07	Metabolic	-	0.505	0.41609476	0.59872715
ADH7	3.862	6.39E-12	4.79E-09	Metabolic	-	0.276	0.6040843	0.75100198
UGT1A10	3.797	3.52E-07	1.04E-05	Metabolic	-	0.485	0.47654616	0.65090706
UGT1A3	3.692	1.17E-07	4.08E-06	Metabolic	-	0.577	0.36608899	0.55313354
ALDH3A1	3.574	8.64E-11	1.62E-08	Metabolic	-	0.005	0.99301751	0.99646932
WFDC2	2.446	7.43E-07	1.71E-05	Stemness	-	0.579	0.20650854	0.38417833
PEG10	1.791	0.00024981	0.00148696	Stemness	-	-0.487	0.27289587	0.46017491
SFRP1	1.781	0.00012949	0.00087493	Stemness	-	0.090	0.83101981	0.90348777
LGR6	1.731	4.19E-05	0.00035815	Stemness	-	-0.037	0.92251786	0.95710801
VWA2	1.678	1.18E-07	4.44E-06	Stemness	-	0.001	0.99749075	0.99871802

We added one sentence in Methods Line 723-725 to clarify the reason why we adopted k=4, “Our choice of the optimal cluster count in NMF was based on the combination factors of NMF cophenetic coefficient and the heterogeneous molecular signatures sufficiently revealed by our data.”.

2. Line 110-111 Fig. 1b does not show “interferon gamma pathway” or “chemokine signaling pathway”. Were these pathways significantly downregulated in the stemness subgroup?

Response: We appreciate the reviewer’s comment here (in line 112-114). We’ve included the two terms “interferon gamma pathway” or “chemokine signaling pathway” into the Figure 1b heatmap, In Figure 1b, the heatmap was based on the ‘-log10’ transformed p-values from the hypergeometric test against the mSigDB database genesets (v.6.2) using signature genes from each transcriptomic group (Extended Data Table S2), while the enrichment plots of three immune pathways (downregulated in stemness, Figure 1b) were based on the GSEA analysis (<https://www.gsea-msigdb.org/gsea/index.jsp>) developed by the Broad Institute, using all profiled genes and associated ranked t-statistics. Both methods independently highlighted that all immune pathways were significantly upregulated in the immunogenic subgroup, but downregulated in the stemness subgroups (revealed by GSEA), and genes from immune pathways were not over-represented in the stemness signature genes (by the hypergeometric test).

3. Line 115-120 The authors claimed histological characteristics of each subtype. However, there is little evidence to support their claim because they just showed the representative images and did not show any quantitative data extracted from histological images.

Response: We appreciate the reviewer’s insightful comment. This was also pointed by reviewer #2 comment #2. To further quantify and compare the histopathological features of four subgroups of ESCC, we developed a deep-learning model using five state-of-the-art convolutional networks, namely Inception-V3, Inception-ResNet-V2, DenseNet-121, VGG16 and ResNet-50, and performed feature extraction on selected tiles from each whole slide image (WSI) from the scanned H&E sections . We then compared features among the four transcriptomic groups, and identified features

strongly associated with each group (i.e., features that were significantly higher in one group compared to the rest). We then selected top five features for each group, and summed these features to create four stable, statistically significant histological markers of gene expression-based classifications, namely DIFF-Feature, MET-Feature, IMM-Feature and STEM-Feature. We performed this analysis using a 65%:35% discovery and validation/test of our WSI samples, see Methods Line 834 – 867 for full detail, also Extended Data Figure 1c. Tiles with the highest scores of each feature were selected for reviewing by a pathologist (Figure 1c).

Indeed, tiles with the highest subtype specific features all contained the distinct histopathological features corresponding to each molecular subgroup (Figure 1C). DIFF-Feature, characterised by that keratin pearls within tumours and differentiated squamous dysplastic cells were significantly higher in the differentiated subgroup samples than the non-differentiated samples (Figure 1c and Extended Data Figure 1c). The MET-Feature, marked by moderate differentiated squamous tumour cells, with some tumour cells showing eosinophilic cytoplasm with less immune cell infiltration. The immunogenic subgroup showed extensive immune cell infiltration within tumours compared to other groups; The stemness subgroup presented poorly differentiated tumour cells and had few immune cell infiltrating within the tumour (Figure 1c and Extended Data Figure 1c). We have included the AI digital pathology results into the revised manuscript Line 152-169. We also include a paragraph discussing this in Line 489 – 500.

4. Line 124-127 The result is unreliable because functional analysis of SFPR1 overexpression was performed in just one cell line. Also, it is highly recommended to show the result of the knockdown experiment using more than one effective shRNA to exclude off-target effects.

Response: According to the reviewer's comment, SFPR1 overexpression experiment was performed in another cell line KYSE-140, and more effective shRNAs were designed to confirm the knockdown results. All shRNA sequences were selected from Sigma which were validated by the provider. The new results are presented in Extended Data Figure 4, and shRNA sequence were provided in the methods section Line 930-940.

5. Line 154 The association between "stemness" and "NK cell estimates" is not strong because the correlation r-value of 0.32 – 0.4 is not high.

Response: We agree with the reviewer's comment, and have changed our wording throughout the manuscript, and state "suggesting a degree of certain association between stemness and NK cell estimates." in Line 205-206.

6. Line 165-166 The authors showed immunohistochemical images of Sample 401 in Fig.2g and Extended Data Fig. 5c. How is the difference of LGR6-staining in the same tumor explained? Also, the architecture of cancer cells is different. Are these derived from the same tumor sample?

Response: Thanks for providing us with the valuable comment. We apologize for any inconvenience caused by the mix-up of samples depicted in Fig 2g. (now Fig. 2h) Upon further investigation, we have confirmed that the staining of CD160 and LGR6 shown in original Fig 2g are from sample 341, but due to our limited perspective, its sequential segmentation may not have been apparent. We have provided additional supporting evidence of this rebuttal (Supplementary Figure 1, shown below) indicating that both are indeed from the same source. To provide greater clarity, we have replaced the staining in Figure 2g for sample 341 with two new field of views that are more similar, as illustrated in the new Figure 2h.

Supplementary Figure 1. Representative immunohistochemical staining image of the tumor tissue stained with anti-CD160 and anti-LGR6 for sample 341.

We have also updated Extended Data Fig 6c accordingly, and showed the anti-CD160 and anti-LGR6 staining for Sample 369, and anti-XCL1 and anti-LGR6 staining for Sample 401 in sequential sections. All these confirmed the co-expression of LGR6 with XCL1 or CD160 in a subset of our samples. We have updated this section of results in Line 212 – 219.

7. Line 198-201 The result is incredible because the authors performed overexpression of XCL1 in just one cell line. In addition, it is strongly recommended to show the result of the knockdown experiment of XCL1 to support their claim.

Response: Following this reviewer's insightful comment, we have further performed the overexpression of XCL1 in two additional XCL1-low cell lines, KYSE-180 and KYSE-410, and the results were consistent to what was shown in XCL1-low KYSE-150 cell line, overexpression of XCL1 significantly increased the IC50 values, making them more resistant to 5-FU, the new results have been added into (Figure 3g). Knockdown of XCL1 would be ideal to consolidate the conclusion, but we could not make XCL1 significantly knockdown subclone cell lines, possibly due to the low proportion of XCL1 overexpressing cells in ESCC cell lines (from our IHC data and sRNA deep sequencing data, XCL1 positive cells are less than 1% of total tumour cells). The alternative approach CRISPR Cas9 knockout strategy was also considered, but it is difficult to design XCL1-specific gRNA given the the high similarity to XCL2. However, we think the overexpression of XCL1 in two additional cell lines is strong enough to support our claim that the overexpressing of XCL1 in cancer cells is associated with the sensitivity to 5-FU.

8. Line 247-250 Among EP300 mutations, 30 % were truncating mutations or deletion. The authors showed higher expression of the mutated allele. However, if the cancer had a truncating/deletion mutation, the mutated allele is expected to result in loss of function. Therefore, their claim of a gain-of-function mode of EP300 mutations is not supported.

Response: We thank the reviewer for this useful comment. We further looked into the allelic imbalance for the eight samples with mutations in EP300, including 4 missense, 1 splice site, 1

nonsense, 1 frame-shift deletion and 1 in-frame deletion, and compared variant allele frequency (VAF) between the WES DNA and RNA-seq data. Three single nucleotide mutations (2 missense mutations, c.4312T>C, c.4355C>G and one splice site mutation, c.4617+1G>A) showed significantly elevated levels of the alternative (mutated) allele in RNA- compared to the DNA-level, as shown in current Fig. 4f. Amongst the remaining five mutations, one nonsense (c.3244C>T) and one frame-shift deletion (c.1914_1915del) in fact showed significantly decreased levels of the alternative allele in RNA than in DNA, suggesting potentially loss-of-function for the mutated allele. We have included this data as Supplementary Table S9, and updated the wording accordingly (Line 314-319), as “Three single nucleotide mutations (2 missense mutations, c.4312T>C, c.4355C>G and one splice site mutation, c.4617+1G>A) showed significantly elevated levels of the alternative (mutated) allele in RNA- compared to the DNA-level. While for two codon-affecting mutations, one nonsense (c.3244C>T) and one frame-shift deletion (c.1914_1915del), the opposite pattern was observed, with decreased levels of the alternative allele in RNA than in DNA”. However, the result for higher EP300 expression levels in EP300-mutated patient samples than in EP300-wildtype samples (Fig. 4e) remains unchanged on the RNA expression level comparison.

9. Line 254-277 The authors showed a higher frequency of NOTCH1 mutation as well as EP300 mutation in the stemness subgroup. Therefore, these results may derive from the effect of NOTCH1 mutation. To investigate the effect of EP300 mutation, it is reasonable to compare EP300-wild and mutant cancers among the same subgroup because the genetic backgrounds of subgroups are different from each other.

Response: We thank the reviewer for this comment. However, as shown in Fig. 4a, the mutational landscape figure among the four transcriptomic groups, mutations in EP300 and NOTCH1 occurred almost exclusively. Only one patient sample that had a missense mutation in EP300, also had a missense mutation in NOTCH1 (fourth sample in the Stemness panel), with another sample having a missense mutation in EP300 and copy number deletion in NOTCH1 (second sample in the Stemness panel). Thus, it is less likely that what we observed between EP300 mutant versus wildtype samples was driven from the effect of NOTCH1 mutations. In fact, we selected NOTCH1 mutated samples (excluding samples with copy number changes in NOTCH1) and compared them with NOTCH1 wildtype samples by differential expression analysis and GSEA. The results demonstrated that the dysregulated genes and pathways were completely different between EP300 mutant vs. wildtype and NOTCH1 mutant vs. wildtype (Supplementary Figure 2a-b of this rebuttal below). The most significantly upregulated pathways (GSEA results) for EP300 mutant vs. wildtype included mitotic prometaphase, mRNA processing and splicing, ATM and Fanconi pathway, spliceosome, DNA replication and chromosome maintenance, while signalling by BMP was the only upregulated pathway for NOTCH1 mutant vs. wildtype. For the most significantly downregulated pathways, the EP300 mutant had core matrisome, integrin1 pathway, cytokine-cytokine receptor interaction, ECM glycoproteins and a large number of immune pathways downregulated compared to wildtype, while the NOTCH1 mutant had RNA POL1 promoter opening, packaging of telomere ends, amyloids, telomere maintenance, meiotic synapsis and recombination and meiosis as top downregulated pathways compared to wildtype. In addition, there was no correlation in the global transcriptomic changes between EP300 mutant vs. wildtype and NOTCH1 mutant vs. wildtype (Supplementary Figure 2c, see below). Furthermore, there was no correlation in mRNA expression between EP300 and NOTCH1 (Supplementary Figure 2d). Therefore, all the results here suggest that our results observed for EP300 mutations were not derived from the effect of NOTCH1 mutations. We have addressed this in the revised manuscript following this reasonable comment.

Supplementary Figure 2. Comparison of differentially expressed genes and pathways between EP300 mutant vs. wildtype and NOTCH1 mutant vs. wildtype. A, Overlap of significantly differentially expressed genes (limma $p < 0.01$) between EP300 mut vs. wt and NOTCH1 mut vs. wt. B, Up- and downregulated pathways (GSEA FDR $q < 0.05$) and the comparisons between EP300 (mut vs. wt) and NOTCH1 (mut vs. wt). C, Scatter plot of log2 fold changes for all the profiled genes between EP300 (mut vs. wt) and NOTCH1 (mut vs. wt). D, Scatter plot of mRNA expression between EP300 and NOTCH1.

Of note, it is not possible to perform meaningful DE analysis comparing EP300 mutant vs wildtype within the same subgroup, as the number of mutated samples was too low in each subgroup, $n=1$ in differentiated, $n=2$ in immunogenic, $n=0$ in metabolic and $n=5$ in stemness groups.

10. Line296-299 In Fig. 9c left panel, three subtypes other than the stemness subtype showed a comparable frequency of truncating mutations in wnt pathway genes. Therefore, because the prediction of the effect of mutations is more reliable for truncating mutation than for missense mutation, it seems that deleterious Wnt pathway mutations are similar.

Response: We appreciated the reviewer’s comment here. Our investigation here focused on comparing the effect of mutations in Wnt pathways genes between stemness and other subtypes. We added one sentence in Line 364-365 to make it clearer, “The stemness subtype also seemed to have the highest proportion of nonsense mutations in Wnt pathway genes”. For the SIFT and PolyPhen-2 analysis, we further clarified this as it was for “missense” mutations in Line 373.

Minor points:

Line 99-101 They designated four subgroups based on known functional gene expression rather than GSEA.

Response: We updated the wording here to “Functional annotation of representative genes in each cluster against known gene sets annotated these subtypes as....” In Line 102

Reviewer #2 (Remarks to the Author): Expert in tumour immunology, NK cells, and oesophageal cancer

Comments for the manuscript entitled “Stemness and NK-like signatures define the poorest subtype

of oesophageal squamous cell carcinoma” by Guozhong Jian et al.

This referee aims to comment on the non-bioinformatical part of this manuscript, leaving the comments for the precise bioinformatics to experts on bioinformatics and statistics.

The manuscript presents an impressive molecular investigation of oesophageal squamous cell carcinoma (ESCC) from 120 Chinese patients. The authors have performed transcriptomics with GSEA and immune profiling, and mutational enrichment analyses, with the aim to find subtypes of tumors, and link this information to novel immune evasion strategies or drug sensitivity. The focus of the study is using bioinformatics, and the results from the authors own patient cohort is compared to previous studies on related cohorts, especially regarding the immune profiling strategy. There is no doubt that the study is of large importance, with ESCCs being one of the deadliest cancers worldwide today, with no effective therapies at hand. A study of this size, at this detailed molecular level, is difficult to find in the literature and will make a difference. Whereas the bioinformatics part of this study seems strong, the immunology and pathology sections are less robust. The manuscript is written in a very condensed scientific English, which makes it difficult to actually understand what has been done in the various panels of experiments. This also makes the reader insecure of what the authors actually have accomplished (the materials and methods (and figure legends) does not help or guide). To be able to fully understand this study, the manuscript has to be rewritten in a more detailed manner. Some comments are given below.

Response: We are glad that the reviewer found our study interesting and important for the field. We also thank the reviewer for such constructive comments for consolidating our conclusions to improve the quality of this manuscript. We have rewritten the manuscript in many places and added more detail in materials and methods, figure legends for various panels of experiments, which further clarified our results and improved the robustness.

1) Every panel of each figure needs to be explained better, both experimentally and statistically, to be able to understand what has been done. This goes for the material and methods part too. Controls should be added where appropriate, and in cases where a value is calculated compared to a control, the control needs to be described equally well. Full statistics should be explained also in Figure legends.

Response: Many thanks for these careful comments, we have extensively addressed and revised our manuscript in all aspects raised by this reviewer. We have expanded and explained our figure panels, figure legends and results.

2) Row 114. “The molecular subtypes present distinguished histological features”. Is this based on these four photographs of four tumors? How representative are they? Where are the statistics behind this to be able to state this quite important finding? The authors should have access to all tumors, and thus be able to perform a tumor tissue array of paraffinized samples, to compare the histology of all 120 tumors on one single glass slide using a light microscope. This way statistics can be added.

Response: This is the same comment as comment #3 of reviewer #1. Our statement of the molecular subtypes present distinguished histological features was based on our observation of whole tumour tissue sections. In order to quantitatively and comprehensively identify the features of each subtype, we had generated the digitised whole slide images of H&E histology slides for all the tumour tissue, we instead developed a deep-learning/AI model, which used five pre-trained models, namely Inception-V3, Inception-ResNet-V2, DenseNet-121, VGG16 and ResNet-50, and performed feature extraction and correlated these features with the four transcriptomic subtypes to

identify significantly enriched histopathological features for each subgroup (see detail in the response of comment #3 of reviewer #1). These AI results still support our previous statement, with more quantitative measures to distinct molecular subtypes of ESCC based on the simple histopathological features. We believe this AI approach assessing the whole slide images of the tissue offers the more comprehensive overview of the tumour tissue compared to tissue array suggested here. Our deep-learning model identified significant enriched histopathological features in each transcriptomic subtype, which showed great consistency to the subtype annotation derived from signature genes of each subtype.

3) Row 116-120. The authors describe these histological features using nomenclature of keratin pearls, extensive immune infiltration etc, without showing it or even presenting how this was done or annotated. Again, to say this a tumor tissue array would be good and should be annotated and evaluated statistically. The material and methods section should be updated accordingly, presenting how keratin pearls or immune cell infiltration was assayed.

Response: We thank the reviewer for this insightful comment. Please see above response the AI model. We have also added the AI model part in the material and methods, and results. Furthermore, regarding immune infiltration, we also performed IHC of various immune cell markers, including CD4, CD8, and CD56, quantified and compared the IHC measurements among the transcriptomic and immune subtypes, in Fig2.c and Extend Data Figure 8a-b. The results demonstrated that immunogenetic subtype and the C2 subtype indeed had the highest level of immune cell infiltration, which was also supported by our AI pathology results (Extend Data Figure 1c). Regarding the quantification of keratin pearls, our deep-learning resource identified the DIFF-Feature significantly higher in the differentiated group compared to other groups. Tiles with high DIFF-Feature scores were enriched for keratin pearls (Figure 1c). we have added this part of the results in Line 152-169.

4) Row 126. Xenograft models are presented in the text, but no ethical permit or precise description of how it was exactly performed in present in the text or materials and methods. Just very vague descriptions referring to Extended data Fig 3. What is Ext data Fig 3a? Explain each panel. Extended data Fig 3 is a typical representative for the concerns mentioned above of too limited explanations.

Response: Thanks for the reviewer's comment. The ethical permit and details about Xenografts model were added in the Methods line 938-940 in the revised manuscript. Besides, more results, descriptions and figure legends for new Extended data Fig. 4 (previously Extended data Fig. 3) were also added in the revised manuscript.

5) Figure 2e, row 149, is this a Fisher's exact test? Where is the statistics shown in the figure? The figure legend is minimal.

Response: This is Fisher's exact test we used to test if there is any difference in the proportion of transcriptomic subtypes between different immune subtypes. We have added the statistics in the figure and explained in detail in the figure legend, Fig. 2f (line 605-607).

6) Fig 2g. IHC controls are absent. A normal tissue should be used as control, and a cell line lacking (negative) or expressing (positive) should be shown to trust the IHC antibody specificity. Again, a tissue tumor array would be good to be able to conclude the results, only four samples are not enough to conclude.

Response: Thanks for the reviewer's helpful suggestion. All negative control had been performed before conducting IHC assay, and these results was now provided in Extended data Fig. 7 and Fig. 9.

Besides, all our staining were performed in paired slides including adjacent normal and cancer tissues, these results have been added in Extended data Fig 9. In fact, we had stained some markers in ESCC tissues microarrays (Shanghai Xinchao Biotechnology Co., Ltd.), while due to the sample area was too small and mostly devoid of submucosal adjunction tissue, so we finally used 99 pairs of original pathological sections for staining analysis of XCL1 and detailed results were provided in the revised manuscript Line 212-219, Supplementary Table S4 and S5 of the revised manuscript to summarise the IHC staining of LGR6, XCL1 and CD160 in consecutive sections. In addition, the antibodies specificity were also confirmed by protein atlas or other users, and our Western blot results.

7) Row 158-166. How is this IHC annotation performed and shown? The authors show statistics in the text that is not described in detail, and cannot be found in the figures or legends. Is it Pearson Chi-square tests or what fo authors mean with co-expression? Tables of annotations? This needs to be clearly shown.

Response: We thank the reviewer for this insightful comment. We did IHC staining for detection of XCL1, CD160 and LGR6 in serial setions of each specimen. The H-score was calculated for the expression of LGR6 and XCL1 by the following formula $H\text{-score value} = [1 \times (\% \text{ cells } 1+) + 2 \times (\% \text{ cells } 2+) + 3 \times (\% \text{ cells } 3+)]$ in QuPath program. The number of cases coexpressing LGR6 and XCL1 or LGR6 and CD160 were counted and the percentage was calculated. We have included the detailed annotation of LGR6 and XCL1 IHC staining for all 99 available samples and CD160 IHC staining for 91 samples in the new Supplementary Table S4. Based on this table, we further counted the cases with LGR6+/XCL1+, LGR6+/XCL1-, LGR6-/XCL1+, LGR6-/XCL1- in the format of 2x2 contingency table to test if these two markers were expressed co-occurently or not, using the two-tailed Fisher's exact test (Supplementary Table S5). This procedure was also done for the LGR6 and CD160 staining (Supplementary Table S4-5). Our IHC results show that LGR6 and XCL1 staining were significantly associated (co-stained or co-expressed) in our samples (two-tailed Fisher's exact test, $p < 0.0001$), but LGR6 and CD160 staining were not significantly associated ($p = 1$).

We have updated this section of results in Line 212-219.

8) From row 177 and on "NK-like tumor cells" are used to described tumor cells expressing CD160. It would be preferable to not name the tumor cells "NK-like tumor cells", since they are tumor cells and not lymphocytes.

Response: We agree with the reviewer's suggestion, and have changed the wording to "tumour cells expressing NK marker genes", and removed "NK-like tumour cells" from our manuscript (From line 171). We also changed "NK-like signatures" to "NK marker XCL1 related signatures" to be more explicit in the manuscript.

9) Fig 3f. How is the IC50 experiment performed exactly. The materials and methods do not describe what is being shown in panel 3f. How are the numbers calculated? At what concentration, time and compared to what? If a Mann-Whitney test is used, what exact control are the bars compared to?

Response: Following the reviewer's comments, we had revised and drug sensitivity analysis in the Method part Line 881-897 "Drug sensitivity analysis". In the new Figure 3g (previous Fig. 3f), The IC50 experiment was performed on KYSE-150, KYSE-180, KYSE-410 as normal controls and the corresponding XCL1 overexpressed cells from the same three cell lines. Mann-Whitney test is used to compare the IC50 values between the control vectors and the corresponding XCL1 overexpressing cells. To make it clear we now modified the figure labels: "control" changed to "vector", "XCL+" changed to "XCL^{OE}".

10) Figure 4 present the genomic mutational landscape. This is important. The authors continue with XCL1 as a promising biomarker for drug sensitivity, using cell lines, and EP300 for survival in the patient cohort. How can these be linked in patient material and survival?

Response: We thank the reviewer for this comment. We further investigated the expression of XCL1 in our patient cohort, and found that high XCL1 expression was significantly associated with worse overall survival (Extended Data Fig 10c). This is consistent with our results of NK cells and their clinical association (Figure 2e), since XCL1 is one of the major markers of our NK cell in silico estimates. Furthermore, we also found that both XCL1 and XCL2 had much more elevated expression in patient tumour samples compared to their matched normal (Extended Data Fig 10a-b). All these data suggest that XCL1 may have a tumour promoting role in ESCC. We have added these in the manuscript line 230-234.

Furthermore, as demonstrated in Fig. 4b ESCC cell line mutational landscape, XCL1 high vs. low, 36% of XCL1 high cell lines had mutations in EP300, while there is no EP300 alteration in XCL1 down-regulated cell line. This observation supports the potential association between XCL1 overexpression and EP300 mutations in the ESCC cancer cells.

11) Figure 5, the authors present immune infiltration and association with clonal diversity, and is also important. It would be good to perform IHC panels on some basic immune populations of the tumors to be able to support the data from the Shannon diversity index. Immune exclusion of T cells compared to M2 macrophages, as well as NK cells would be interesting to see. If CD160 is expressed on tumor cells.

Response: We have performed IHC on immune cell markers of CD4, CD8, CD56 and compared IHC measurement among the subtypes, in Fig 2c and Extend Data Figure 8. Consistent with our RNA-seq results, the IHC results demonstrated that the levels of these immune cells were significantly higher in immunogenic and C2 subgroups compared to other groups (Extended Data Fig 8 and Fig 2c). We also correlated the IHC measurements of immune cells with the Shannon diversity index (SDI) measured from the genomic data, and found the significantly negative correlations between them for CD8+ (correlation coefficient, $r=-0.29$, $p=0.017$) and CD56+ ($r=-0.274$, $p=0.017$) stained cells (Extended Data Fig 8c), supporting our results of the potential interplay between immune infiltration and tumour clonal evolution (i.e., higher immune infiltration is associated with lower level of ITH, measured by Shannon diversity) (Fig. 5f). We have added this result in Extended Data Figure 8, and in Line 378 – 380.

Regarding the expression of CD160, the mRNA expression of CD160 was the highest in C3 immune subgroup, but not in C2 subgroup (Extended Data Fig 6a), and the IHC of CD56 and CD160 (Extend Data Figure 7 and 9) further demonstrated that CD160 was not expressed by CD56+ NK cells but by tumour and dysplastic cells. Indeed, we observed the evidence of CD160 expressed by LGR6+ tumour cells in 25 out of 91 cases (27%) (Supplementary Table S5). We included the co-staining data of CD56, CD160 and LGR6 in consecutive sections in Extended Data Figure 9. However, across the whole cohort of 91 available samples, LGR6 and CD160 IHC staining were not significantly associated ($p= 1$, i.e., not significantly co-expressed by the tumour cells across the cohort).

(12) The manuscript needs to be rewritten, to add much more detail for each panel, to be able to understand what has been done. As it is now, it is difficult to interpret the results.

Response: We have followed the reviewer's advice and added much more detail in the results, figure legends and Methods and Materials for various panels of experiments throughout the manuscript.

Reviewer #3 (Remarks to the Author): Expert in cancer stem cells

In this study Jiang et al., aims at defining novel molecular subgroups in human Oesophageal Squamous Cell Carcinoma (ESCC) by analysing the transcriptome and genome of a big cohort of ESCC from Chinese patients.

The authors, using bulk-RNA sequencing, identify four distinct ESCC transcriptomic subtypes. Among them one shows the worst survival rate and is characterized by the expression of WFDC2, SFRP1, LGR6 and VWA2 genes ("stemness" subtype). The data obtained from the RNA sequencing analysis is then analysed using deconvolution methods to determine the tumor immune composition of the ESCC samples. Using this approach, they divide the tumors in three categories according to the content/type of immune cells present within the sample and identify a group of tumors showing high expression of NK-like gene signature, the group associated with the poorest survival rate. Interestingly, they observe that tumors showing high expression of XCL1 (gene upregulated within the NK gene signature and C3 immune type) present higher resistance to several drugs. Thirdly, the authors characterize the genomic landscape of the same cohort of patients and describe genomic landscapes similar to those described in the numerous genomic analyses that have been reported on ESCC (The Cancer Genome Atlas Research, N. et al. Nature 541, 169–175 (2017); Song, Y. et al. Nature 509, 91–95 (2014); Lin, D. C. et al. Nat. Genet. 46, 467–473 (2014); Gao, Y. B. et al. Nat. Genet. 46, 1097–1102 (2014)). Finally, the authors determine if there is any correlation between the transcriptomic and immune signatures and the mutational landscape of the ESCC analysed.

The main criticism is that this study lacks accuracy as many of the statements are not well sustained or only based on correlations. There is a lack of validation of the transcriptomic and immune signatures in biological samples and lack of experiments in biological samples aimed at supporting the numerous correlations present in the study. These validations and experiments would certainly help to strengthen and confirm the message of the paper.

Remarkably, there are some points that need to be further developed or clarified:

1-Using bulk RNA-sequencing the authors classify the ESCC analysed in 4 different subtypes characterized by the expression of different markers, subtypes that show different morphological characteristics and patient survival rates (Figure 1). The authors claim that they are interested to characterize both the tumor and tumor microenvironment. However, they do not analyse if the genes found to be upregulated in the different subgroups are expressed by the tumor cells or by the tumor microenvironment, which is extremely relevant to validate and strengthen this key point the authors want to make. This validation could be accomplished by performing immunostainings for the different markers in tumors from the 4 ESCC subtypes or alternatively using techniques that would allow a deeper characterization of the different cell populations present in the samples (single cell RNA sequencing). Using the latter approach, a recent paper (Zhang, X., Peng, L., Luo, Y. et al. Nat Commun 12, 5291, 2021) identified eight different common expression programs of epithelial cells in ESCC tumors and characterized the populations present in the tumor microenvironment of the same tumors.

Response: We thank the reviewer for such constructive comments here. We utilised the single cell RNA-seq data of 208,659 single cells from 60 individuals of ESCC (Zhang et al. Nat Commun 12, 5291,

2021) to validate our subtype specific gene signatures. Based on the annotated cell clusters, we were able to see in which cell types / subtypes our signature genes were expressed. Out of the 208,659 cells, 44,122 were epithelial cells that were dominantly cancer cells (Zhang et al. Nat Commun. 2021). As shown in our Fig. 1a and Extended Data Figure 3, signature genes from differentiated, metabolic and stemness subtypes were mainly expressed by epithelial cells. Moreover, signature genes from these three subtypes seemed to be expressed by different subpopulations of epithelial cells, i.e., occupying different epithelial subclusters in UMAP. In fact, our differentiated subtype corresponds to the epithelial differentiation (Epi1/2) program identified by Zhang et al., with the overlap of many signature genes, such as LGALS7, LGALS7B, KRT16, KRT6B/C, FABP5 and LY6D of the Epi1 programme, S100A7/8/9, SPRR1A/B and SPRR2D of the Epi2 programme. Our metabolic subtype corresponds to the oxidative stress or detoxification (Oxd) program, with shared genes as CES1, ALDH1A1, ALDH3A1, AKR1C1/2/3 and GPX2 of the Oxd program. Although our stemness subtype was not characterised by the eight epithelial gene signatures of Zhang et al., the feature plot of LGR6, WFDC2 and SFRP1 in single cells clearly showed that these genes were expressed by a distinct subpopulation of epithelial cells (Fig. 1a and Extended Data Figure 3). As expected, signature genes of our immunogenic subtype were all expressed by non-tumour cells. For example, MS4A1, CD79A and MZB1 were expressed by B cells, CXCL9 was expressed mainly by myeloid cells, with some in fibroblasts, endothelial and pericytes (Fig. 1a and Extended Data Figure 3). Therefore, the single cell results further validated our findings derived from bulk tissue RNA-seq, and supported our four distinct transcriptomic subtypes. We have included this set of results in Line 118-137. In addition, we have also done IHC for several immune cells characterisation, it is indeed that our four molecular subtypes show different immune cells infiltration (see our response to Reviewer 2 comment 11).

2- In Figure 2 the authors identify a cluster, cluster 3, characterized by the expression of NK markers, again it is not clear if these NK markers are expressed by the tumor cells (staining for XCL1 and Lgr6 Fig2 g superior panel is too weak to be able to conclude anything and morphologically the tumor shown does not seem a “stemness” tumor) or if there is infiltration of NK cells or presence of NK cells surrounding the tumors (staining for a NK marker would be extremely helpful). Therefore the description of the immune infiltration and immune microenvironment is superficial and needs to be improved. In addition, only 50% of the stemness-subtype tumors could be classified as belonging to the immune cluster 3 (Fig 2e), so the statement “given the strong association between stemness and NK cell estimates- line 154-155” is overstated and should be corrected.

Response: We thank the reviewers for the insightful comment here. Our RNA-seq data showed that NK markers XCL1, XCL2 and CD160 were the highest in the C3 cluster (Fig 2b and Extended Data Fig6a). Following the reviewer’s advice, we further stained NK cell marker CD56 in our cohort, and the IHC results showed that the level of CD56 was the highest in C2 cluster, but significantly lower in both C1 and C3 clusters (Fig 2c). This was also supported by the mRNA expression of other NK markers, such as NKG7 and KLRC1, being the highest in C2 cluster, but low in C3 and C1 clusters (Extended Data Fig 6a). These results suggest that the level of CD56+ NK cell infiltration was low in the C3 cluster. Following further inspection of the XCL1 and LGR6 IHC staining in stemness and other groups, we concluded that XCL1 were not expressed by NK cells (CD56+) in our tissues, instead by tumour cells and other dysplastic epithelial cells of submucosa glands. These results suggest that XCL1 was expressed by tumour cells and dysplastic cells of submucosa gland across our cohort. We have included the IHC staining results in Figure 2h, Extended Data Fig 6-9, Supplementary Table S4-5, and added the results in Line 212-219.

We also agree with the reviewer's comment regarding "the strong association between stemness and NK cell estimates", and have changed our wording throughout the manuscript to "there is a degree of positive correlation between stemness and NK estimate high subgroups".

3- One of the genes that characterize Cluster 3 (NK-like signature) is XCL1. The authors show using GSEA that XCL1-high cells exhibit upregulation of drug metabolism of cytochrome P450, retinol metabolism and biological oxidations (line 185) and in line 106 the authors state "the metabolic subtype is associated with the upregulation of genes involved in drug metabolism by cytochrome P450 and retinol metabolism". Would this mean that tumors showing high XCL1 expression belong to the "metabolic subtype"? This is confusing and highlights that the link between XCL1 (NK-like) and "stemness subtype" or "metabolic subtype" should be better explained.

Response: We thank the reviewer for this detailed observation. To further test how similar the pathway activity is for drug metabolism by cytochrome P450 and retinol metabolism between XCL1-high vs. low and metabolic vs. others comparisons, we compared the t-statistics derived from limma differential expression analysis (a statistics combining both log2 fold change 'magnitude' and p-value 'significance') for genes involved in these two pathways. As shown in the figures below (Supplementary Figure S3 of this rebuttal), there was a weak correlation in t-statistics between XCL1-high and metabolic subtypes for genes involved in drug metabolism by cytochrome P450 and retinol metabolism. This analysis suggests that different subsets of genes contributed to the upregulation of these two pathways seen in XCL-high cells and the metabolic subtype. Furthermore, there was also very minimum number of signature genes shared between XCL1-high and metabolic subtype. Therefore, there is no evidence that suggests tumours showing high XCL1 expression belong to the metabolic subtype.

Supplementary Figure S3

The association between immune and transcriptomic subtypes was based on Fig. 2f. Out of 31 samples in C3 (NK high), 15 were from the stemness, 6 from metabolic, 8 from differentiated and 2 from immunogenic subtypes, suggesting there is a degree of enrichment of stemness samples in the C3 subgroup. We have followed the reviewer's advice, and rephrase the association between stemness and NK estimate high subgroups throughout the manuscript. We have also added a sentence in Line 248-250, "it is worth noting that different subsets of genes contributed to the upregulation of drug metabolism by cytochrome P450 and retinol metabolism were seen in XCL-high cells and the metabolic subtype" to further clarify the point here.

4- In Figure 2g, Extended Data Figure 5c and line 164-172, the authors state that XCL1 is exclusively expressed in cancer cells showing adenocarcinoma morphology. Should we consider all XCL1-expressing tumours adeno-squamous cell carcinomas? If so, the authors should repeat the analysis

by studying the ESCC and adeno-squamous cell carcinomas separately as relevant differences may be found in terms of transcriptomics, genomics and response to chemotherapy.

Response: We appreciate the reviewer's insightful comment here. Following this reviewer's comments, we comprehensively reviewed the whole section of resected samples of IHC staining for XCL1 expression from 99 available samples, XCL1 expression was predominantly expressed in cancer cells showing adenocarcinoma morphology, but some squamous carcinoma cells also expressed XCL1. Therefore, we have updated our description and conclusion in our revised manuscript accordingly. Esophageal adeno-squamous cell carcinoma is a very rare disease clinically. It is an extremely difficult task to assemble an Esophageal adeno-squamous cell carcinoma sample cohort in a decent size that allows to investigate the transcriptomic / genomic landscape with enough statistical power. To study the transcriptomics, genomics and response to chemotherapy of adeno-squamous cell carcinomas is an extremely interesting and valuable topic scientifically and clinically. However, this is beyond the scope of our current study.

5- The authors suggest that XCL1-high cells are slow cycling (line 191-193). This is an important point that could explain the higher chemotherapy resistance observed (Fig3). I would recommend the authors to study this point in more detail as it is quite relevant and could increase the impact of the study. BrdU/EdU incorporation studies can help to uncover this matter.

Response: We thank the reviewer for this comment. To further support our results based on RNA-seq, we utilised the single cell RNA-seq data from Zhang et al. (Nat Commun 12, 5291, 2021) to perform further studies. Focussing on the epithelial cells, we were able to identify 515 XCL1 positive cells (expression level CPM>0) and 32,944 XCL1 negative cells (expression level =0). We then calculated the expression level of each gene programme of cell cycle on single cell level (e.g., REACTOME_CELL_CYCLE_CHECKPOINTS, REACTOME_CELL_CYCLE_MITOTIC) using Seurat AddModuleScore function, and showed that XCL1 positive (XCL1+) cells had significantly lower cell cycle activities than XCL1 negative (XCL1-) cells (Wilcoxon test, $p=7.48e-06$ for REACTOME_CELL_CYCLE_CHECKPOINTS; $p=0.001$ for REACTOME_CELL_CYCLE_MITOTIC), which is consistent to our data. We have added these results in Fig. 3e and Line 256-259. Of note, overexpression of XCL1 does not significantly affect the cell proliferation (see extended data Fig 11d), the reviewer's suggestion to do BrdU/EdU incorporation might be helpful, which is to be performed in a separate project.

REVIEWER COMMENTS

Reviewer #1 (Remarks to the Author):

The authors have substantially clarified the figures and manuscript, and addressed the majority of points raised, but still the first point remained.

In the Results section (lines 97-), they claimed that they first performed unsupervised clustering and then identified gene signatures based on the classification. However, in their rebuttal, they explained that they chose $k=4$ (the number of clusters) based on NMF cophenetic coefficient and heterogeneous molecular signatures. As these molecular signatures appear to be predefined, they intentionally opted for four clusters, which represent differentiation, immunogenicity, metabolic characteristics, and stemness. If this is indeed the case, they should have initially mentioned the prior knowledge of the four classifications.

Reviewer #2 (Remarks to the Author):

The authors have answered my comments and improved the clarity/statistics in the text.

Reviewer #3 (Remarks to the Author):

The data and figures have improved substantially. However, in this revision the authors have only partially addressed my previous questions and comments. The new version of manuscript is difficult to follow as it is written in a rather condensed manner and the supplementary figures do not follow the same order as the text (ie. Extended Data Fig 6b).

There are some points that need to be addressed:

1. In Figure 1a lower panel the authors analyse the scRNAseq data from another study (Zhang et al). However, it is not possible to interpret this figure 1a without going to Ext DataFig 3. The main figures of an article should be self-explanatory. For this reason, the description of the clusters should be in Figure 1a not in a supplementary figure. Or alternatively the authors should move all the scRNAseq to Ext Data Fig 3.

2. In Figure 1c the authors should better describe and annotate within the images the features characteristic of each of the subtypes. And ideally, validate the results of the bulk RNA seq using in situ hybridization or immunofluorescence.

3. The quality of many of the stainings presented are poor.

Fig 2 and Ext Data Fig 9: stainings for XCL1, CD56 and CD160 not to work. In Ext Data Fig 9 XCL1 and the rest of the stainings are not “matched serial sections of tumour tissues” as described in the text line 208. The staining of serial sections for the markers described is essential to prove the authors’ hypothesis and should be performed accurately.

The images for LGR6 staining need to be improved.

4. In line 136 the authors claim “Therefore, the single cell results further validated our findings derived from bulk tissue RNA-seq, and supported our four distinct transcriptomic subtypes”. In the scRNAseq data from Zhang et al the stemness subtype is not described, so this statement should be corrected.

5. In line 248 the authors claim “ It is worth noting that different subsets of genes that contributed to the upregulation of drug metabolism of cytochrome P450 and retinol metabolism were seen in XCL-high cells and metabolic subtype”. How did the authors reach that conclusion? What and where is the data that made them conclude that? This point should be better explained.

6. The authors did not reply to my previous question number 5:

The authors suggest that XCL1-high cells are slow cycling (line 191-193). This is an important point that could explain the higher chemotherapy resistance observed (Fig3). I would recommend the authors to study this point in more detail as it is quite relevant and could increase the impact of the study. BrdU/EdU incorporation studies can help to uncover this matter.

The answer that they provide is a correlation based on the data of the scRNA seq from Zang et al. and from that correlation it cannot be concluded that the cells are slow-cycling/ quiescent. Performing BrdU/EdU incorporation studies in cell lines or staining for quiescence markers in human samples is relatively quick to accomplish and needed in order to conclude such a statement.

8. In the abstract line 50 the authors state “functional mutation enrichment in the Wnt signalling”. What does this mean: mutations that lead to the in/activation of the Wnt signalling pathway? This should be clarified.

Minor comment:

The image scale in Ext Data Fig7 needs to be improved.

Reviewer #4 (Remarks to the Author): Expert in oesophageal cancer subtypes, genomics, and bioinformatics

Disclaimer: I did not participate in the first round of review of the initial submission, and I am providing my review on their revised work.

In this manuscript, the authors performed whole-exome sequencing and bulk RNA-seq on 120 Chinese ESCC patients. They identified 4 RNA-defined subtypes: differentiated, metabolic, immunogenic and stemness. Using deep-learning model, they showed that these 4 subtypes were associated with different histological features, implying their biological basis. Moreover, they identified that NK cell markers including XCL1 and CD160, were surprisingly expressed by a subset of tumor cells. XCL1 were followed upon and its expression confers the sensitivity of ESCC cells to chemotherapeutic drugs.

This is a comprehensive work of tour de force: there were large amount of sequencing data, bioinformatic analyses, massive imaging data processing, as well as fairly extensive biological investigations. Some of the findings are strong and may have significant implications, such as the identification of 4 subtypes and XCL1 investigation. However, I do find a number of weaknesses which need to be addressed.

Major issues:

- 1) What is the relationship between these newly designated subtypes with the TCGA subtypes? This can be done by looking at shared pathways enriched in different subtyping schema. For example, in TCGA, ESCC1 has frequent upregulation of oxidative pathways. Is ESCC1 similar as the Metabolic subtype here?
- 2) The validation using external single-cell RNA-seq data is useful but somewhat superficial. What was shown was merely that the selected genes from different subtypes could be expressed by different clusters from scRNA-seq, which is largely expected. What really needs to be addressed is do these different clusters from scRNA-seq show similar biology or transcriptional programs to bulk RNA-seq defined subtypes?
- 3) The investigation of SFRP1 needs further development. The authors described that IHC staining was done on its protein, but did not mention how many tumor and normal samples were stained. Is SFRP1 expressed higher in tumor vs normal samples? What is the percentage of SFRP1 expression in tumor samples? Since it was proposed as a stemness related genes, does it correlate with the differentiation status of the tumors? The knockdown efficiency (Fig.s1e,f) appears poor, and the WB needs to be quantified.
- 4) Associating histological features to different transcriptomic subtypes is helpful and can support their biological underpinnings. However, at this scale (Fig.1C), the histological features are difficult to discern for most readers who are not trained pathologists. For example, it is hard to see immune cells or eosinophilic cytoplasm at this resolution. It would be very helpful to add certain IHC markers to make these histological features conspicuous.

5) The investigation of XCL1 in cancer cells was a highlight in this work. However, I could not find any IHC image showing strong XCL1 staining. The ones included, such as Fig.2h, Fig.s6c, were very weak. The authors need to show convincing strong IHC signals of XCL1 in cancer cells, since this is one of their key findings.

6) Line 224-228. "Interestingly, we observed that XCL1-expressing dysplastic cells in the submucosal gland in most of the cases are completely separated from the squamous cell carcinoma, suggesting that this subgroup of patients might concurrently have both squamous cell carcinoma and adenocarcinoma or this adeno-squamous carcinoma might be derived from submucosa gland epithelial cells". These are indeed very interesting! However, no data was shown. Can the authors perform a statistical analysis to compare the histology of these tumors vs. XCL1 expression?

7) Line 256-258, it is a bit concerning that XCL1 is only expressed in 515 out of 33459 (1.5%) total epithelial cells from the single-cell RNA-seq data. Can the authors explain why such low frequency? Is it because of the high cutoff used? This low positivity negatively affects the significance of the biological contribution of XCL1 in ESCC.

Minor points:

1) The RNA-seq and WES data are still under controlled access and not publicly available. I think it is Nat Communication's policy that these data will need to be made publicly available upon acceptance.

2) "an independent cohort of 65 ESCC patients" in Fig.1e was not described in the methods or results. Where were these samples from? Were they all primary tumors?

Reviewer #5 (Remarks to the Author): Expert in digital pathology and deep learning

In this article, the authors propose a new subclassification of esophageal squamous cell carcinoma. This is clinically interesting because most previous efforts at subtyping this disease also included other histologies of esophageal cancer such as adenocarcinomas or gastroesophageal junction tumors. It is sensible to focus exclusively on squamous cell esophageal cancer and attempt to subtype it.

My expertise is specifically in digital pathology, so I am reviewing the digital histopathology model which the authors utilized.

Unfortunately, there are some concerns regarding the deep learning analysis.

1. Some methodological details are not entirely clear. For instance, what does a downsampling factor of 64 fold mean? The authors should specify their final resolution in micrometers per pixel and then provide the absolute pixel size for their tiles.

2. Additionally, they mention that they manually discarded some images. This is concerning, and it is crucial for them to clearly state why these images were discarded. What precisely constituted "poor quality"? Moreover, it would be beneficial to know how many images per class were discarded. The

manuscript would benefit from a CONSORT-style diagram to elucidate these points.

3. It is also vital that the study adheres to the STARD (or similar) guidelines, ensuring that all boxes in this guideline are ticked.

4. Regarding the design of the algorithm, the authors mention that they extracted features with five pretrained neural networks. However, the rationale for this decision is unclear. This choice, which appears quite uncommon, needs further explanation. Typically, one would use just a single robust network, selected through a hyperparameter tuning experiment on a dedicated tuning set. Alternatively, one might opt for a network that has been validated in previous studies. Ideally, the contemporary standard involves not using a network pre-trained on ImageNet, but rather one that has been pre-trained in a self-supervised manner on histopathology images, such as the RetCCL network or the CTransPath network. In summary, the image analysis algorithm presented seems non-standard. While the authors might have had valid reasons for these unconventional design choices, they need to elucidate their reasoning in the manuscript. It would be even more beneficial if the authors employed a state-of-the-art pipeline (such as CLAM or the one from Wagner et al., *Cancer Cell*, 2023), or if they presented benchmarking experimental results showing their methods' superiority compared to more conventional methods.

5. Lastly, there are concerns regarding the statistical measures for the gene expression classification. The authors used a t-test, but they did not specify whether all the prerequisites for a t-test were met. This information needs to be explicitly stated.

Point by point responses to the Reviewers comments

Reviewer #1 (Remarks to the Author):

The authors have substantially clarified the figures and manuscript, and addressed the majority of points raised, but still the first point remained.

In the Results section (lines 97-), they claimed that they first performed unsupervised clustering and then identified gene signatures based on the classification. However, in their rebuttal, they explained that they chose $k=4$ (the number of clusters) based on NMF cophenetic coefficient and heterogeneous molecular signatures. As these molecular signatures appear to be predefined, they intentionally opted for four clusters, which represent differentiation, immunogenicity, metabolic characteristics, and stemness. If this is indeed the case, they should have initially mentioned the prior knowledge of the four classifications.

Response: Thanks for this reviewer's appreciation on our substantial work done in the revised manuscript. We did not predefine any molecular signatures before the NMF clustering. NMF clustering is an unsupervised clustering algorithm that identifies patient clusters simply based on the expression values of input genes, i.e., in our case, top 1,500 variable genes with largest mean absolute deviation values across samples were used. Although $k=2$ gave the best cophenetic correlation $r=0.99$, this was followed by almost equally great performances of $k=4$ and $k=3$, with both having the cophenetic correlation $r>0.98$ (Extended Data Fig. 1a). As $k=4$ uncovered the level of heterogeneity and granularity of transcriptomic patterns in a finer resolution, and samples were evenly distributed among the four groups, $k=4$ was therefore selected. This has led to the discovery of our four transcriptomic clusters.

We have updated the wording in Methods as "Although $k=2$ gave the best cophenetic correlation $r=0.99$, $k=4$ also achieved great clustering performance with cophenetic correlation $r=0.985$, and the latter ($k=4$) uncovered the level of heterogeneity and granularity of transcriptomic patterns in a finer resolution, thus was selected for our investigation", highlighted in line 742-745.

Reviewer #2 (Remarks to the Author):

The authors have answered my comments and improved the clarity/statistics in the text.

Response: We really appreciate Reviewer #2's positive response.

Reviewer #3 (Remarks to the Author):

The data and figures have improved substantially. However, in this revision the authors have only partially addressed my previous questions and comments. The new version of manuscript is difficult to follow as it is written in a rather condensed manner and the supplementary figures do not follow the same order as the text (ie. Extended Data Fig 6b).

There are some points that need to be addressed:

1. In Figure 1a lower panel the authors analyse the scRNAseq data from another study (Zhang et al).

However, it is not possible to interpret this figure 1a without going to Ext Data Fig 3. The main figures of an article should be self-explanatory. For this reason, the description of the clusters should be in Figure 1a not in a supplementary figure. Or alternatively the authors should move all the scRNAseq to Ext Data Fig 3.

Response: Following the reviewer's helpful comment, we have moved all single cell RNA-seq related data to the new Ext Data Fig 3 and 4, and also created a new Ext Data Fig 4 regarding the subtypes of ESCC epithelial single cells, as requested by the comment #2 of Reviewer #4.

2. In Figure 1c the authors should better describe and annotate within the images the features characteristic of each of the subtypes. And ideally, validate the results of the bulk RNA seq using in situ hybridization or immunofluorescence.

Response: Thanks for this helpful comment. We have redone the Fig 1c with the inserted high magnification pictures to address the specific features of each subtype, in addition to the further explanation in the figure legend. We completely agree with this reviewer's comment, in situ hybridization or immune-fluorescence staining is an ideal way to validate the bulk RNA sequencing data, regrettably we were not able to do this in an available time frame due to the difficulty to find specific probes and antibodies specifically recognising the specific markers in each subtype of ESCC. However, one of the major findings in this manuscript is that we developed AI pathology (see new Extended Data Fig. 6), which can correlate and predict the subtype of ESCC based on the H&E morphology. The features described in Fig 1C with further annotation may be sufficient to support the conclusion of this manuscript. Furthermore, we have also validated our transcriptomic subtypes using the single cell RNA-seq data of 60 ESCC samples (new Ext Data Fig 3 and 4). Therefore, we hope the reviewer would agree that the data and results supported from the deep-learning histopathology model and single-cell RNA-seq were strong and convincing to support the subtypes derived from our bulk RNA-seq.

3. The quality of many of the stainings presented are poor.

Fig 2 and Ext Data Fig 9: stainings for XCL1, CD56 and CD160 not to work. In Ext Data Fig 9 XCL1 and the rest of the stainings are not "matched serial sections of tumour tissues" as described in the text line 208. The staining of serial sections for the markers described is essential to prove the authors' hypothesis and should be performed accurately.

The images for LGR6 staining need to be improved.

Response: Apologises for this. We found that the pictures in the convert version of pdf file for the reviewer's review were indeed not clear due to the compressed file. We have now provided the high quality of all pictures in Fig 2h, Extended Data Fig 8-9 and Extended Data Fig 11 in the new version. In particular, we have redone Ext Data Fig 11 (the previous Ext Data Fig 9) for XCL1 staining using the matched serial sections of tumour tissues and the corresponding normal tissues in our revised manuscript (Ext Data Fig 11 panel a and panel b). In particular, the data demonstrated the co-staining of LGR6 and CD160 and/or XCL1 in many ESCC tumour cells which were CD56 negative.

We sincere hope that this reviewer will be satisfied with the new set of pictures.

4. In line 136 the authors claim "Therefore, the single cell results further validated our findings

derived from bulk tissue RNA-seq, and supported our four distinct transcriptomic subtypes". In the scRNAseq data from Zhang et al the stemness subtype is not described, so this statement should be corrected.

Response: We thank the reviewer for this comment. This comment is highly related to the second comment from Reviewer #4. We have updated this section as follows:

To further look into the heterogeneity and granularity of transcriptional programmes of ESCC epithelial cells, the NMF clustering with k=10 factors was performed on ~44,000 epithelial cells of Zhang et al., Nat Comms 2021 (similar techniques also used in Gojo et al., Cancer Cell 2020, Lai et al., Int J Cancer 2021 and DeMartino et al., Nat Comms 2023, and many others), followed by the differential expression analysis using the Seurat "FindMarkers" function to identify top differentially expressed genes. Signature genes for each cluster were identified based on adjusted p-value <0.0001 and log2 fold change >1. Top 50 signature genes were then selected based on the log2 fold change for each cluster (new Extended Data Fig. 4). The NMF clusters were annotated based on their signature genes and up/down-regulated pathways. Their corresponding expression programmes of Zhang et al., 2021 and our related transcriptomic subtypes were identified based on shared signature genes (Extended Data Fig. 4). The NMF Cluster 5 and 10 appeared to correspond to the differentiated subtype, with Cluster 4 corresponding to the metabolic subtype, Cluster 6 corresponding to the immunogenic subtype, while Cluster 1 shared many stemness signature genes, such as SFRP1, WFDC2 and LGR6 (Extended Data Fig. 4). Cluster 1 also had significantly upregulated Wnt signalling and NCAM1 interactions, and the most downregulated keratinization / cornified envelope and metabolism pathways, which were all signature pathway activities for the stemness subtype. Reassuringly, all previous eight expression programmes of epithelial cells from Zhang et al., 2021 were identified in our single-cell NMF clusters. These single cell results further validated and support the robustness of our four transcriptomic subtypes derived from bulk tissue RNA-seq.

We have updated this section in Results section highlighted in line 125-147 and Methods section line 909 – 916.

5. In line 248 the authors claim " It is worth noting that different subsets of genes that contributed to the upregulation of drug metabolism of cytochrome P450 and retinol metabolism were seen in XCL-high cells and metabolic subtype". How did the authors reach that conclusion? What and where is the data that made them conclude that? This point should be better explained.

Response: We previously showed this result only in our responses to the reviewers' comments, but not included in the main text, please see below (in green font) from the rebuttal of last round. We have now included this figure in the new Extended Data Fig. 13 and the associated results in the figure legend to clarify and support this point here, in new Line 267-270 (Extended Data Fig. 13).

"Response: We thank the reviewer for this detailed observation. To further test how similar the pathway activity is for drug metabolism by cytochrome P450 and retinol metabolism between XCL1-high vs. low and metabolic vs. others comparisons, we compared the t-statistics derived from limma differential expression analysis (a statistics combining both log2 fold change 'magnitude' and p-value 'significance') for genes involved in these two pathways. As shown in the figures below (Supplementary Figure S3 of this rebuttal), there was a weak correlation in t-statistics between XCL1-high and metabolic subtypes for genes involved in drug metabolism by cytochrome P450 and retinol metabolism. This analysis suggests that different subsets of genes contributed to the upregulation of these two pathways seen in XCL-high cells and the metabolic subtype. Furthermore, there was also very minimum number of signature genes (i.e., significantly upregulated genes)

shared between XCL1-high and metabolic subtype. Therefore, there is no evidence that suggests tumours showing high XCL1 expression belong to the metabolic subtype.

Supplementary Figure S3

We have also added a sentence in Line 256-258, “it is worth noting that different subsets of genes contributed to the upregulation of drug metabolism by cytochrome P450 and retinol metabolism were seen in XCL-high cells and the metabolic subtype” to further clarify the point here.”

6. The authors did not reply to my previous question number 5:

The authors suggest that XCL1-high cells are slow cycling (line 191-193). This is an important point that could explain the higher chemotherapy resistance observed (Fig3). I would recommend the authors to study this point in more detail as it is quite relevant and could increase the impact of the study. BrdU/EdU incorporation studies can help to uncover this matter.

The answer that they provide is a correlation based on the data of the scRNA seq from Zang et al. and from that correlation it cannot be concluded that the cells are slow-cycling/ quiescent. Performing BrdU/EdU incorporation studies in cell lines or staining for quiescence markers in human samples is relatively quick to accomplish and needed in order to conclude such a statement.

Response: Following this insightful comment, we did Edu incorporation studies and cell cycle analysis of XCL1 overexpressed cell lines and the counterpart control cell line, the detailed results were shown in the new Extended Data Fig. 14, and also included the method section Line 965-975. It seemed that there was no significant difference of Edu incorporation between the XCL1 overexpressed cell lines and the control cell lines. This suggests that XCL1 did not seem to affect the G1/S phase of cell cycle although mRNA expression levels of cell cycle related genes were significantly reduced in XCL1 high cells compared to low cells. This warrants further investigation. We have rephrased our results of this part as follows in Results Line 278-282 in our revised version.

“Although mRNA expression levels of cell cycle related genes were reduced in XCL1-high cells compared to XCL low cells, XCL1 overexpression in ESCC cells did not functionally affect cell cycle (Extended Data Fig. 14). More work is needed to further elucidate its role in cell cycle and other pathways associated with ESCC.”

In addition, we have rephrased the term of “cell cycle activities” with the term of “cell cycle gene set enrichment scores” to more accurately reflect the results derived from RNA-seq and single-cell RNA-seq gene set enrichment analysis, as shown in Results Line 273-278.

8. In the abstract line 50 the authors state “functional mutation enrichment in the Wnt signalling”. What does this mean: mutations that lead to the in/activation of the Wnt signalling pathway? This should be clarified.

Response: We thank the reviewer for raising this point. This point was previously made clear in our Results “Functional mutation enrichment in pathways among ESCC subtypes” section.

Our results shown in Figure 5d/e and Extended Data Fig. 18 suggest that there were significantly more functionally relevant mutations in the Wnt/ β -catenin signalling genes in the stemness subtype, compared to other subtypes (Results Line 373-377). In Results Line 379-385, “we then evaluated whether the pathway functional mutation enrichment affected the expression of pathway genes, and found that mutation enrichment in Wnt/ β -catenin signalling, inflammatory response and hypoxia were positively correlated with pathway gene expression activities for the corresponding ESCC subtypes (Fig. 5e), suggesting that high level of Wnt signalling expression in the stemness group could be a consequence of functional mutations in regulators of the Wnt pathway, and these functional mutations were the most enriched in stemness samples”. Thus, our results did support that these functionally relevant mutations may lead to the activation of the Wnt pathway. However, we were careful with our wording, as such observation was not validated by further biological function studies, which is beyond the scope of this study.

Minor comment:

The image scale in Ext Data Fig7 needs to be improved.

Response: We have done this in our revised manuscript, now the new Ext Data Fig 9.

Reviewer #4 (Remarks to the Author): Expert in oesophageal cancer subtypes, genomics, and bioinformatics

Disclaimer: I did not participate in the first round of review of the initial submission, and I am providing my review on their revised work.

In this manuscript, the authors performed whole-exome sequencing and bulk RNA-seq on 120 Chinese ESCC patients. They identified 4 RNA-defined subtypes: differentiated, metabolic, immunogenic and stemness. Using deep-learning model, they showed that these 4 subtypes were associated with different histological features, implying their biological basis. Moreover, they identified that NK cell markers including XCL1 and CD160, were surprisingly expressed by a subset of tumor cells. XCL1 were followed upon and its expression confers the sensitivity of ESCC cells to chemotherapeutic drugs.

This is a comprehensive work of tour de force: there were large amount of sequencing data, bioinformatic analyses, massive imaging data processing, as well as fairly extensive biological investigations. Some of the findings are strong and may have significant implications, such as the identification of 4 subtypes and XCL1 investigation. However, I do find a number of weaknesses which need to be addressed.

We thank the reviewer for the positive response of our study.

Major issues:

1) What is the relationship between these newly designated subtypes with the TCGA subtypes? This can be done by looking at shared pathways enriched in different subtyping schema. For example, in TCGA, ESCC1 has frequent upregulation of oxidative pathways. Is ESCC1 similar as the Metabolic subtype here?

Response: We thank the reviewer for this comment. The TCGA ESCC molecular subtypes 1/2/3 were derived from the clustering using multi-omics data including mutations, mRNA/miRNA expression, DNA methylation and copy number aberrations, while our four subtypes were only based on transcriptomics. For the 90 TCGA ESCC samples, they came from patients with very diverse ethnic background, including Brazil (n=15), Canada (n=4), Russia (n=12), Ukraine (n=4), United States (n=14) and Vietnam (n=41), while our 120 samples all came from a high ESCC incidence area (Henan Province) in China. Therefore, it is not feasible to compare the subtypes derived from different omics technologies and different ethnic background. However, we did some analysis of association between our subtypes and TCGA subtypes, as shown in Figure 4A (mutation oncoplot) and addressed in the Discussion, there was no association in mutations profiles between TCGA and our subtypes.

Following the reviewer's advice, we also performed the differential expression analysis and gene set enrichment analysis between TCGA ESCC subtypes using TCGA mRNA expression data. Only ESCC1 (n=50, 56%) had three pathways upregulated at an FDR level < 0.25, including RESPONSE_TO_OXIDIZED_PHOSPHOLIPIDS (FDR = 0.15), and ESCC2 (n=36, 40%) and ESCC3 (n=4, 4%) had no significantly up- or down-regulated pathways. For ESCC1, apart from oxidized phospholipids, no other metabolic and oxidative pathways were significantly upregulated compared to ESCC2/3. Therefore, based on all these data, there did not seem to be strong associations between TCGA and our subtypes, as they were derived from different omics signatures and different ethnic background.

2) The validation using external single-cell RNA-seq data is useful but somewhat superficial. What was shown was merely that the selected genes from different subtypes could be expressed by different clusters from scRNA-seq, which is largely expected. What really needs to be addressed is do these different clusters from scRNA-seq show similar biology or transcriptional programs to bulk RNA-seq defined subtypes?

Response: We really appreciate the reviewer for this insightful comment. To further investigate the heterogeneity and granularity of transcriptional programmes of ESCC epithelial cells, the NMF clustering with k=10 factors was performed on ~44,000 epithelial cells of Zhang et al., Nat Comms 2021 (similar techniques also used in Gojo et al., Cancer Cell 2020, Lai et al., Int J Cancer 2021 and DeMartino et al., Nat Comms 2023), followed by the differential expression analysis using the Seurat "FindMarkers" function to identify top differentially expressed genes, as well as gene set enrichment analysis for each cluster. Signature genes for each cluster were identified based on adjusted p-value <0.0001 and log2 fold change >1. Top 50 signature genes were then selected based on the log2 fold change for each cluster (new Extended Data Fig. 4). The NMF clusters were annotated based on their signature genes and up/down-regulated pathways. Their corresponding expression programmes of Zhang et al., 2021 and our related transcriptomic subtypes were identified based on shared signature genes (Extended Data Fig. 4). The NMF Cluster 5 and 10 appeared to correspond to our differentiated subtype, with Cluster 4 corresponding to the metabolic subtype, Cluster 6 corresponding to the immunogenic subtype, while Cluster 1 shared many stemness signature genes, such as SFRP1, WFDC2 and LGR6 (Extended Data Fig. 4a). Cluster 1 also had significantly upregulated Wnt signalling and NCAM1 interactions, and the mostly downregulated keratinization / cornified envelope and

metabolism pathways, which were all signature pathway activities for the stemness subtype. Reassuringly, all previous eight expression programmes of epithelial cells from Zhang et al., 2021 were identified in our single cell NMF clusters. These single cell results further validated and support the robustness of our four transcriptomic subtypes derived from bulk tissue RNA-seq.

We have updated this section in Results section line 125-147 and Methods section line 909-916.

3) The investigation of SFRP1 needs further development. The authors described that IHC staining was done on its protein, but did not mention how many tumor and normal samples were stained. Is SFRP1 expressed higher in tumor vs normal samples? What is the percentage of SFRP1 expression in tumor samples? Since it was proposed as a stemness related genes, does it correlate with the differentiation status of the tumors? The knockdown efficiency (Fig.s1e,f) appears poor, and the WB needs to be quantified.

Response: SFRP1 was found as one of a set of genes showing the overexpression of mRNA in the Stemness group compared to other three groups based on the differential RNA expression in 120 tumour samples of ESCC. The frequency of SFRP1 protein expression was low in human ESCC, IHC assay revealed SFRP1 protein was positive in 4.3% (3/70) of ESCC tissue and no positive in the matched normal samples. We have provided this detailed data in our revised manuscript, Line 155-158.

Although SFRP1 is a stemness related genes, its expression does not significantly correlate with the differentiation status. The knockdown efficiency in WB is quantified, shown by Ext Data Fig 5 panel g and h, indicating significant 20-40% knockdown efficiency in KYSE-520 cells and ~50% knockdown efficiency in KYSE-450.

4) Associating histological features to different transcriptomic subtypes is helpful and can support their biological underpinnings. However, at this scale (Fig.1C), the histological features are difficult to discern for most readers who are not trained pathologists. For example, it is hard to see immune cells or eosinophilic cytoplasm at this resolution. It would be very helpful to add certain IHC markers to make these histological features conspicuous.

Response: We fully understood this reviewer's comment, similar as the comment #2 from Reviewer 3. We have redone the Fig 1c with the inserted high magnification pictures to address the specific features of each subtype, with more explanation in the text and figure legend. Please see more our response to the comment 2 from the reviewer 3. We have done some IHC for immune cells, such as CD4, CD8 and CD56, which showed significantly higher expression in immunogenic group (see Figure 2c, Extended Data Fig 9 and Fig 10). Given the lack of specific IHC markers to differentiate the histopathological features presented in each subtype in an available time frame, we are not able to provide more IHC staining to make these histopathological features conspicuous as suggested by the reviewer although it would be indeed helpful. However, we have developed a deep-learning AI histology model that strongly correlated histological features with transcriptomic subtypes. This AI model identified subtype specific histological features and these distinct features corresponded nicely with their respective transcriptomic subtypes. Please see the section of "Distinct histological features among transcriptomic subtypes" Line 164-188 for more detail. Furthermore, we have also validated our transcriptomic subtypes using the single cell RNA-seq data of 60 ESCC samples (new Ext Data Fig 3 and 4). Therefore, we hope the reviewer would agree that the data and results supported from the deep-learning histopathology model and single-cell RNA-seq were strong and convincing to support the subtypes derived from our bulk RNA-seq.

5) The investigation of XCL1 in cancer cells was a highlight in this work. However, I could not find any IHC image showing strong XCL1 staining. The ones included, such as Fig.2h, Fig.s6c, were very weak. The authors need to show convincing strong IHC signals of XCL1 in cancer cells, since this is one of their key findings.

Response: Apologies for this. We found that the pictures in the converted version of pdf file for the reviewers were indeed not clear somehow. We now have provided the high quality of all pictures with higher magnification, which demonstrate the real positive staining of XCL1 in cancer cells although the overall staining of XCL1 staining is weaker than other markers, in Fig 2h, new Ext Data Fig 8 and Ext Data Fig 11.

6) Line 224-228. "Interestingly, we observed that XCL1-expressing dysplastic cells in the submucosal gland in most of the cases are completely separated from the squamous cell carcinoma, suggesting that this subgroup of patients might concurrently have both squamous cell carcinoma and adenocarcinoma or this adeno-squamous carcinoma might be derived from submucosa gland epithelial cells". These are indeed very interesting! However, no data was shown. Can the authors perform a statistical analysis to compare the histology of these tumors vs. XCL1 expression?

Response: Really appreciate this reviewer's comment on this interesting finding. Adeno-squamous carcinoma (ASC) of the esophagus is an uncommon type of esophageal cancer that contains both adenocarcinoma and squamous cell carcinoma elements. Data on this biologically unique type of cancer are limited and mainly stem from case reports and small case series. Given that the incidence of adeno-squamous carcinoma of human oesophageal cancer is very low (between 0.37%-1%), the number of ASC in this study is too low to perform a meaningful statistical analysis. We are approaching more collaborators to get more samples for the further investigation on this very interesting finding.

7) Line 256-258, it is a bit concerning that XCL1 is only expressed in 515 out of 33459 (1.5%) total epithelial cells from the single-cell RNA-seq data. Can the authors explain why such low frequency? Is it because of the high cutoff used? This low positivity negatively affects the significance of the biological contribution of XCL1 in ESCC.

Response: We thank the reviewer for raising this comment. XCL1 positive and negative cells were identified based on the Seurat SCTransform based normalised gene-level counts: as positive when normalised gene-level counts >0; and negative when normalised gene-level counts =0. This led to the identification of ~1.5% XCL-positive epithelial cells. We have updated this sentence in Methods Line 905-906 to make it clear.

Low frequency of XCL1 positive cells was also seen in the IHC staining in human primary ESCC samples, the positive cells within whole cancer tissues were around 1-2%. The biological function of XCL1 expression in cancer cells is not fully understood although our preliminary result has demonstrated that XCL1 overexpression decreases the sensitivity of cancer cells to chemotherapy drug. Although XCL1 expresses in a very low proportion of cancer cells, these cells may present the cancer stem cells or therapeutic resistant cells and play some important role in cancer recurrence or poor prognosis. This definitely warrants further investigation.

Minor points:

1) The RNA-seq and WES data are still under controlled access and not publicly available. I think it is Nat Communication's policy that these data will need to be made publicly available upon acceptance.

Response: The data has been publicized at the National Genomics Data Centre of China with the Bioproject Access ID of PRJCA001577 (<https://ngdc.cncb.ac.cn/bioproject/browse/PRJCA001577>), and researchers can access the data after filling in a data request form following the guideline set by the Ministry of Science and Technology of China. If there is any difficulty for any researcher to access the data, we could provide help.

2) "an independent cohort of 65 ESCC patients" in Fig.1e was not described in the methods or results. Where were these samples from? Were they all primary tumors?

Response: We have now provided more information about this 65 ESCC patients in our revised manuscript, in Methods Lind 712-714.

They were all primary tumour, treatment naïve, from Anyang Cancer Hospital, under the approve of ethics committee of Both Anyang Cancer Hospital and The First Affiliated Hospital of Zhengzhou University.

Reviewer #5 (Remarks to the Author): Expert in digital pathology and deep learning

In this article, the authors propose a new subclassification of esophageal squamous cell carcinoma. This is clinically interesting because most previous efforts at subtyping this disease also included other histologies of esophageal cancer such as adenocarcinomas or gastroesophageal junction tumors. It is sensible to focus exclusively on squamous cell esophageal cancer and attempt to subtype it.

My expertise is specifically in digital pathology, so I am reviewing the digital histopathology model which the authors utilized.

Unfortunately, there are some concerns regarding the deep learning analysis.

1. Some methodological details are not entirely clear. For instance, what does a downsampling factor of 64 fold mean? The authors should specify their final resolution in micrometers per pixel and then provide the absolute pixel size for their tiles.

Response: We used the Openslide API for handling our whole slide images (WSI). The API processes the multiresolution images as pyramid levels with each subsequent level representing a down-sampled version of the previous level. There is no image scaling by the library, instead levels are already available in the slide file (Goode et al., Journal of pathology informatics 2013). 64 is the down sample factor we used to determine the image level we extracted from the slide files and all the slides were scanned at 20x objective power at 0.44 μm per pixel resolution. The tiles used for feature extraction were extracted at 300 x 300 pixels for all samples. We have updated the method text to make this clear in Methods Line 858-860, and line 867.

2. Additionally, they mention that they manually discarded some images. This is concerning, and it is crucial for them to clearly state why these images were discarded. What precisely constituted "poor quality"? Moreover, it would be beneficial to know how many images per class were discarded. The manuscript would benefit from a CONSORT-style diagram to elucidate these points.

Response: Each WSI was manually reviewed by a qualified pathologist, and poor quality images were discarded under the direct pathologist's supervision, poor quality of imaging means that, the sections were folded without clear morphology or there were not enough tumour cells presented in the slides obtained from the Department of histopathology. A total of 91 WSIs were retained for the deep-learning analysis, i.e., differentiated group, n=28; immunogenic group, n=27; metabolic group, n=18; stemness group, n=18. We have updated this section in Methods Line 862-867.

3. It is also vital that the study adheres to the STARD (or similar) guidelines, ensuring that all boxes in this guideline are ticked.

Response: We have checked and followed the STARD guidelines, report as much detail as required.

4. Regarding the design of the algorithm, the authors mention that they extracted features with five pretrained neural networks. However, the rationale for this decision is unclear.

This choice, which appears quite uncommon, needs further explanation. Typically, one would use just a single robust network, selected through a hyperparameter tuning experiment on a dedicated tuning set. Alternatively, one might opt for a network that has been validated in previous studies. Ideally, the contemporary standard involves not using a network pre-trained on ImageNet, but rather one that has been pre-trained in a self-supervised manner on histopathology images, such as the RetCCL network or the CTransPath network. In summary, the image analysis algorithm presented seems non-standard. While the authors might have had valid reasons for these unconventional design choices, they need to elucidate their reasoning in the manuscript. It would be even more beneficial if the authors employed a state-of-the-art pipeline (such as CLAM or the one from Wagner et al., Cancer Cell, 2023), or if they presented benchmarking experimental results showing their methods' superiority compared to more conventional methods.

Response: We acknowledge this reviewer's point regarding the number of pretrained models used for feature extraction. Utilising multiple pretrained networks allowed us to capture a broader range of features and patterns. Each of the pretrained models utilised different model architectures, which can result in diverse feature representations. For example, the Inception and ResNet architecture are known to have different feature representation due to the use of residual connection in the latter. This diversity can enhance performance by capturing different predictive elements and building more enriched feature representations into our system. As shown in the new Supplementary Table 4, features from different pretrained models contributed differently to the top features for each subtype.

Previous studies, such as Fu et al., Nat Cancer 2020, and Courtiol et al., Nat Med 2019, have efficiently utilised Imagenet pretrained models for extraction of histology features. Although some of these studies opt for either training a new 2D convolutional layer to perform weighted sum between the extracted features (Courtiol et al., 2019) or model fine-tuning, this is not applicable in our study because of our sample size (n=91 slides, distributed as DIFF = 28, IMM = 27, MET = 18 and STEM = 18). These numbers are not suitable for robust model training but will work for classical statistics

comparing the extracted features between defined groups. Our aim was to implement a simple robust system to consistently extract histological features that capture the gene expression subtypes in both Discovery and Test sets. Indeed, we were able to effectively classify the gene expression subtypes using the meta histological features (new Supplementary Table 4 and 5).

Finally, we are aware of the alternative systems such as CLAM and currently use them in our ongoing projects when appropriate. However, these systems require additional model training and often have specific domain unlike the general-purpose image recognition capabilities of the Imagenet pretrained models (Courtiol et al., 2019).

In the revised manuscript, we have elaborated on these points to provide a clearer rationale for using multiple pretrained models for feature extraction. We also added one sentence in Results Line 171-173, "This model diversity can enhance performance by capturing different predictive elements and building more enriched representations into the system." to elaborate the rationale more. We have also included a workflow diagram to clearly describe our steps (new Extend Data Fig 6a). We hope this will address your concern and improve the overall clarity of our process.

Fu Y, Jung AW, Torne RV, Gonzalez S, Vöhringer H, Shmatko A, Yates LR, Jimenez-Linan M, Moore L, Gerstung M. Pan-cancer computational histopathology reveals mutations, tumor composition and prognosis. *Nat Cancer*. 2020 Aug;1(8):800-810. doi: 10.1038/s43018-020-0085-8. Epub 2020 Jul 27. PMID: 35122049.

Courtioi P, Maussion C, Moarii M, Pronier E, Pilcer S, Sefta M, Manceron P, Toldo S, Zaslavskiy M, Le Stang N, Girard N, Elemento O, Nicholson AG, Blay JY, Galateau-Sallé F, Wainrib G, Clozel T. Deep learning-based classification of mesothelioma improves prediction of patient outcome. *Nat Med*. 2019 Oct;25(10):1519-1525. doi: 10.1038/s41591-019-0583-3. Epub 2019 Oct 7. PMID: 31591589.

5. Lastly, there are concerns regarding the statistical measures for the gene expression classification. The authors used a t-test, but they did not specify whether all the prerequisites for a t-test were met. This information needs to be explicitly stated.

Response: Thank you for raising this point. Assumption checks across the extracted features revealed that normality was not met for several of the features and the combined features. Therefore, we have replaced the t-test with the Wilcoxon rank-sum test and repeated the relevant analysis (new Supplementary Table 5, also Methods Line 883). Importantly, our results revealed that these features remain predictive of the subtypes and our interpretation remains the same.

REVIEWER COMMENTS

Reviewer #1 (Remarks to the Author):

The authors have addressed this reviewer's remaining point and revised their manuscript accordingly.

Reviewer #3 (Remarks to the Author):

The authors have substantially addressed the points raised and improved the quality of the text and figures.

Reviewer #4 (Remarks to the Author):

The authors have addressed all my concerns. I have no more critiques.

Reviewer #5 (Remarks to the Author):

The authors have responded to my comments -- thank you

the authors provide some justification for their design choices for the neural network architecture, but they do not solve the issue that the model design is very uncommon and probably not optimal. The fundamental problem seems to be the rather low sample size ($N=91$).

Having said that, the authors are transparent about their approach and although it is not state of the art and there is a risk for overfitting, the methods are transparently reported and the interpretation seems to be backed up by the data.

Major issue: the github repos seem to be unusable (only partial codes stored in the repository, no documentation, poor structure, no example data). With the current github repos it is not possible to reproduce the results.

I suggest to remove the "The" from the title.

Reviewer #5 (Remarks on code availability):

Major issue: the github repos seem to be unusable (only partial codes stored in the repository, no documentation, poor structure, no example data). With the current github repos it is not possible to reproduce the results.

Point by point responses to the Reviewer #5 (Remarks to the Author):

The authors have responded to my comments -- thank you

the authors provide some justification for their design choices for the neural network architecture, but they do not solve the issue that the model design is very uncommon and probably not optimal. The fundamental problem seems to be the rather low sample size (N=91).

Having said that, the authors are transparent about their approach and although it is not state of the art and there is a risk for overfitting, the methods are transparently reported and the interpretation seems to be backed up by the data.

Response: We appreciate that this reviewer agrees that our methods are transparently reported, and our interpretation is backed up by the data.

The ensemble approach of using multiple CNN models for feature extraction is not uncommon and has been used previously although not much in the field of computational pathology, exemplified by

- F. Shaheen and B. Verma, "An ensemble of deep learning architectures for automatic feature extraction," 2016 IEEE Symposium Series on Computational Intelligence (SSCI), Athens, Greece, 2016, pp. 1-5, doi: 10.1109/SSCI.2016.7850047.
- Bhandi, V., Sumithra Devi, K.A. (2021). Feature Extraction from Ensemble of Deep CNN Model for Image Retrieval Application. In: Jeena Jacob, I., Kolandapalayam Shanmugam, S., Piramuthu, S., Falkowski-Gilski, P. (eds) Data Intelligence and Cognitive Informatics. Algorithms for Intelligent Systems. Springer, Singapore. https://doi.org/10.1007/978-981-15-8530-2_57
- Dechao Chen, Yang Chen, Jieming Ma, Cheng Cheng, Xuefeng Xi, Run Zhu, Zhiming Cui, "An Ensemble Deep Neural Network for Footprint Image Retrieval Based on Transfer Learning", Journal of Sensors, vol. 2021, Article ID 6631029, 9 pages, 2021. <https://doi.org/10.1155/2021/6631029>

[Figure Redacted]

All these studies showed that the ensemble approach retains more features for the final retrieval and achieves better accuracy than a separate feature extraction (Shaheen and Verma 2016 IEEE Symposium Series on Computational Intelligence (SSCI); Chen et al., 2021 Journal of Sensors).

Given our low sample size (n=91) of four subtypes, our feature extraction could benefit from this ensemble approach by increasing the numbers of useful features for the final retrieval. The feature representations of the images extracted by different models are different, but these different feature representations contain some essential information components (Chen et al., 2021 Journal of Sensors). Indeed, our results showed that more diverse representative features could be identified using the feature extraction from multiple CNN models (shown in Supplementary Table 4). This produced more robust features that were fused together to create subtype specific Meta Features (Supplementary Table 5).

We have added one sentence in Results Line 173 to highlight the advantage of this ensemble approach, “This ensemble approach was reported to retain more informative features for the final retrieval and achieve better accuracy than a separate feature extraction”, and cited the papers above.

Larger independent cohorts are needed to further validate our ensemble model and top features, which are under planning but beyond the scope of this study.

Major issue: the github repos seem to be unusable (only partial codes stored in the repository, no documentation, poor structure, no example data). With the current github repos it is not possible to reproduce the results.

Response: Following on this comment, We have now added extensive detail about the functions and scripts used in our workflow at <https://github.com/BioInforCore-BCI/giExtract>, forked from <https://github.com/caanene1/giExtract>. We also provide example dataset and run functions inside dedicated folder. In particular, we created a “manuscript” folder in the repos, containing “run.sh”, “downstream.R” and “example data” folder, ensuring the reproducibility of our results.

To make it easy for other researchers to use our code, we have now uploaded the python part of the workflow to the Pypi package database. This enables a user to install the tool like any other python package. The interface provides detailed documentation of the inputs and requirements (see below). Please also visit the Pypi website for our package at <https://pypi.org/project/giExtract/1.0.1/>

```
Last login: Wed Mar  6 18:10:10 on ttys000
chinedus-MacBook-Pro:~ chineduanene$ giExtract -h
usage: giExtract [-h] -p PATH -c CONTEXT -k COLUMN [-b BATCH] [-s NATCOM]

optional arguments:
  -h, --help            show this help message and exit
  -p PATH, --path PATH  Path to images
  -c CONTEXT, --context CONTEXT
                        Context file with at least a column for flowing the images
  -k COLUMN, --column COLUMN
                        The column name in context to use for flowing data
  -b BATCH, --batch BATCH
                        Batch size for flowing the data into the models
  -s NATCOM, --natcom NATCOM
                        Indicate if the NatCom et al features should be calculated
chinedus-MacBook-Pro:~ chineduanene$ █
```

I suggest to remove the "The" from the title.

Response: We have updated the title to “Integrated molecular and histological analysis defines subtypes of esophageal squamous cell carcinoma”

We’d like to take this opportunity to thank all the reviewers for their invaluable comments and critique that has made our study much more thorough and robust!

REVIEWERS' COMMENTS

Reviewer #5 (Remarks to the Author):

Thank you for addressing all of my comments.